DOI: 10.1038/s41467-017-01514-3　　**OPEN**

# Evidence of renal angiomyolipoma neoplastic stem cells arising from renal epithelial cells

Ana Filipa Gonçalves[1,2], Mojca Adlesic[1,3], Simone Brandt[2], Tomas Hejhal[1], Sabine Harlander[1], Lukas Sommer[4], Olga Shakhova[4,5], Peter J. Wild[2] & Ian J. Frew[1,3,6,7]

Renal angiomyolipomas (AML) contain an admixture of clonal tumour cells with features of several different mesenchymal lineages, implying the existence of an unidentified AML neoplastic stem cell. Biallelic inactivation of *TSC2* or *TSC1* is believed to represent the driving event in these tumours. Here we show that TSC2 knockdown transforms senescence-resistant cultured mouse and human renal epithelial cells into neoplastic stem cells that serially propagate renal AML-like tumours in mice. mTOR inhibitory therapy of mouse AML allografts mimics the clinical responses of human renal AMLs. Deletion of *Tsc1* in mouse renal epithelia causes differentiation in vivo into cells expressing characteristic AML markers. Human renal AML and a renal AML cell line express proximal tubule markers. We describe the first mouse models of renal AML and provide evidence that these mesenchymal tumours originate from renal proximal tubule epithelial cells, uncovering an unexpected pathological differentiation plasticity of the proximal tubule.

---

[1] Institute of Physiology, University of Zurich, 8057 Zurich, Switzerland. [2] Department of Pathology and Molecular Pathology, University Hospital Zurich, 8091 Zurich, Switzerland. [3] Department of Hematology, Oncology and Stem Cell Transplantation, Faculty of Medicine, Medical Center - University of Freiburg, 79106 Freiburg, Germany. [4] Institute of Anatomy, University of Zurich, 8057 Zurich, Switzerland. [5] Clinic of Oncology, University Hospital Zurich, 8091 Zurich, Switzerland. [6] Zurich Center for Integrative Human Physiology, University of Zurich, 8057 Zurich, Switzerland. [7] BIOSS Center for Biological Signalling Studies, University of Freiburg, 79104 Freiburg, Germany. Correspondence and requests for materials should be addressed to I.J.F. (email: ian.frew@uniklinik-freiburg.de)

Renal angiomyolipomas (AML) are benign, but life-threatening neoplasms that contain variable admixtures of tumour cells that are histologically and molecularly similar to vascular (angio-), smooth muscle (myo-) and fat (lipo-) lineages[1]. Genetic analyses have shown that these different cell types within an individual AML are clonal[2,3], indicating that they must be derived from a common tumour-initiating cell that has the capacity to differentiate into these different lineages, suggestive of a putative neoplastic stem cell. However, the identities and characteristics of the normal cell of origin of AML and of the presumptive AML stem cell remain unknown. Renal AML cells also express molecular markers of the melanocyte lineage[4,5], which serve as clinical diagnostic markers. As embryonic neural crest stem cells or neural crest-derived progenitor cells from the adult skin can differentiate to form melanocytes, adipocytes and smooth muscle cells[6–8], the cell type of origin of renal AML has been proposed to be an unidentified kidney-resident, neural crest-derived lineage[5]. Others have suggested that myoblasts[9], pericytes[10] or lymphatic endothelium[11] represent the AML cell of origin. It is also possible that another renal cell type could become transformed and reprogrammed to form a neoplastic AML stem cell. In this context is noteworthy that a rare variant of AML, called AMLEC (AML with epithelial cysts) contains epithelial tubular or cystic structures[12]. While it has not yet been conclusively proven, there is some evidence that these epithelial structures may be tumour-derived[13,14], implying that the putative AML or AMLEC neoplastic stem cell may also have the capacity to differentiate into renal epithelial cells in some cases.

Multiple and bilateral renal AMLs develop in up to 80% of patients with the autosomal dominant tuberous sclerosis complex (TSC) syndrome (also known as Bourneville–Pringle disease), affecting ~1 in 6,000 newborns[4]. AMLs represent the most common cause of mortality in adult TSC patients due to spontaneous haemorrhage of abnormal tumour vasculature and can also cause significant morbidity by compressing adjacent normal kidney tissue, thereby impairing kidney function. TSC patients inherit a loss of function mutation in one allele of either of the TSC1 or TSC2 genes and AMLs in these patients almost always display somatic mutation of the wild type allele[15–17]. Roughly 80% of all renal AMLs arise sporadically in the general population, affecting ~0.6% of females and 0.3% of males[18]. These tumours also almost invariably display biallelic loss of TSC2 or more rarely of TSC1[15–17,19]. The TSC1 (also known as HAMARTIN) and TSC2 (also known as TUBERIN) proteins form a complex that negatively regulates the mTORC1 protein kinase complex, a key node in cellular signalling networks that control cellular growth and proliferation. Consistent with the oncogenic role of constitutive mTORC1 activation, everolimus, an inhibitor of mTORC1, causes regression of renal AMLs in many patients[20] and this therapy isapproved by the U.S. Food and Drug Administration and the European Medicines Agency for the treatment of TSC-associated renal AML. Despite the clear role of TSC1 and TSC2 mutation in the human disease, renal AMLs surprisingly did not develop in numerous mouse models involving homozygous or heterozygous deletion of Tsc1 or Tsc2 (reviewed in ref.[21]). However, inducible, ubiquitous Cre-mediated deletion of Tsc1 caused the development of small kidney lesions displaying several characteristic renal AML markers such as HMB45, Smooth Muscle Actin (SMA), CATHEPSIN K and VIMENTIN[22], suggesting that these lesions might represent renal AML precursor lesions. The cell type that gave rise to these lesions was not determined.

In this study we utilised a reverse tumour-engineering approach in mouse and human cells to explore the cell of origin of renal AML. Surprisingly we identified that our renal AML tumour models derive from renal epithelial cells. We further showed that some cells in human renal AMLs, as well as a renal AML cell line, exhibit molecular features of renal proximal tubular epithelial cells. These findings argue that renal AMLs might derive from proximal tubule epithelial cells.

## Results

**Tsc2 and Cdkn2a loss converts renal cells into AML-forming cells.** As renal AMLs are relatively genetically simple tumours that are characterised almost exclusively by recurrent TSC2 or TSC1 mutations[19], we reasoned that it might be possible to reverse engineer renal AML starting from normal primary cells. We first utilised primary mouse embryo fibroblasts (MEFs) to establish RNAi tools to knockdown Tsc1 and Tsc2. Infection with lentiviruses expressing shRNA-Tsc1 or shRNA-Tsc2 efficiently reduced TSC1 or TSC2 protein abundance, increased phosphorylation of ribosomal protein S6 (Ser240/244) and 4E-BP1 (Thr37/46), indicative of mTORC1 activation, and caused accumulation of HIF-1α and the HIF-1α inducible protein GLUT1 (Supplementary Fig. 1a), mimicking previously published effects of knockout of Tsc1 or Tsc2 in MEFs[23–25]. Tsc1 or Tsc2 knockdown also inhibited cellular proliferation and induced premature senescence (Supplementary Fig. 1b, c), another known consequence of loss of function of TSC1 or TSC2[24,25].

As the cell type of origin of renal AML is unknown, we prepared primary cultures from collagenase-digested whole mouse kidneys with the aim of simultaneously isolating different types of kidney cells and cultured them in renal epithelial cell medium. These cultures predominantly comprised E-CADHERIN-expressing epithelial cells with rare VIMENTIN-expressing cells (Supplementary Fig. 2a, b). Knockdown of Tsc1 or Tsc2 in these cultures also caused rapid loss of proliferative capacity (data not shown). Knockdown of TSC2 in primary human renal proximal tubular epithelial cells similarly caused inhibition of proliferation and the flattened, spread cellular morphology that is characteristic of cellular senescence (data not shown). This loss of proliferative capacity, a common consequence of expression of oncogenes or loss of tumour suppressor genes in primary cell culture systems, represents a limitation to an ex vivo tumour engineering approach.

To overcome this problem we identified that Cdkn2a knockout in MEFs rescued senescence and induced cellular transformation upon knockdown of Tsc1 or Tsc2 (Supplementary Fig. 1d–i). Using this information we first observed that while cultures from wild type kidneys lost epithelial appearance upon passaging, cultures from Cdkn2a$^{fl/fl}$ mice infected with a Cre-expressing adenoviral vector to delete Cdkn2a (hereafter designated Cdkn2a$^{Δ/Δ}$ cells) maintained epithelial appearance, proliferated in culture (Fig. 1a) and formed colonies when plated at low density on plastic but did not form colonies in soft agar (Fig. 1a and Supplementary Fig. 2d). Infection of cultured Cdkn2a$^{Δ/Δ}$ renal cells with lentiviruses expressing shRNA-Tsc1 or shRNA-Tsc2 increased the rate of proliferation (Fig. 1a) and allowed the formation of colonies in soft-agar (Fig. 1a and Supplementary Fig. 2d), indicative of cellular transformation. Similarly, infection of cultured renal cells from wild type mice with a lentiviral vector expressing both shRNA-Tsc2 and shRNA-Cdkn2a (shRNA-Tsc2 + Cdkn2a) (Supplementary Fig. 2c) increased proliferation rate compared with shRNA-Cdkn2a knockdown and allowed growth in soft agar (Fig. 1b and Supplementary Fig. 2e). Tsc2 knockdown or Cdkn2a knockdown or knockout alone did not cause cellular transformation (Fig. 1a, b), demonstrating the cooperativity of this genetic effect. Genotypes that grew in soft agar acquired the stem cell-like characteristic of growth as non-adherent spheres that could be serially passaged (Fig. 1c). Approximately 9% of cells from the total virus-infected cellular populations formed

spheres. Importantly, when injected subcutaneously into female *SCID-beige* mice, shRNA-*Tsc2/Cdkn2a$^{\Delta/\Delta}$* ($n = 12$ injections) and shRNA-*Tsc2* + *Cdkn2a* ($n = 16$ injections) sphere-selected cells formed slowly growing tumours within 1–2 months (Fig. 1d).

Tumours did not grow in male mice, potentially consistent with the increased incidence of human renal AMLs in females compared to males[18]. Allograft tumours showed several histological similarities to human renal AML and comprised a mixture

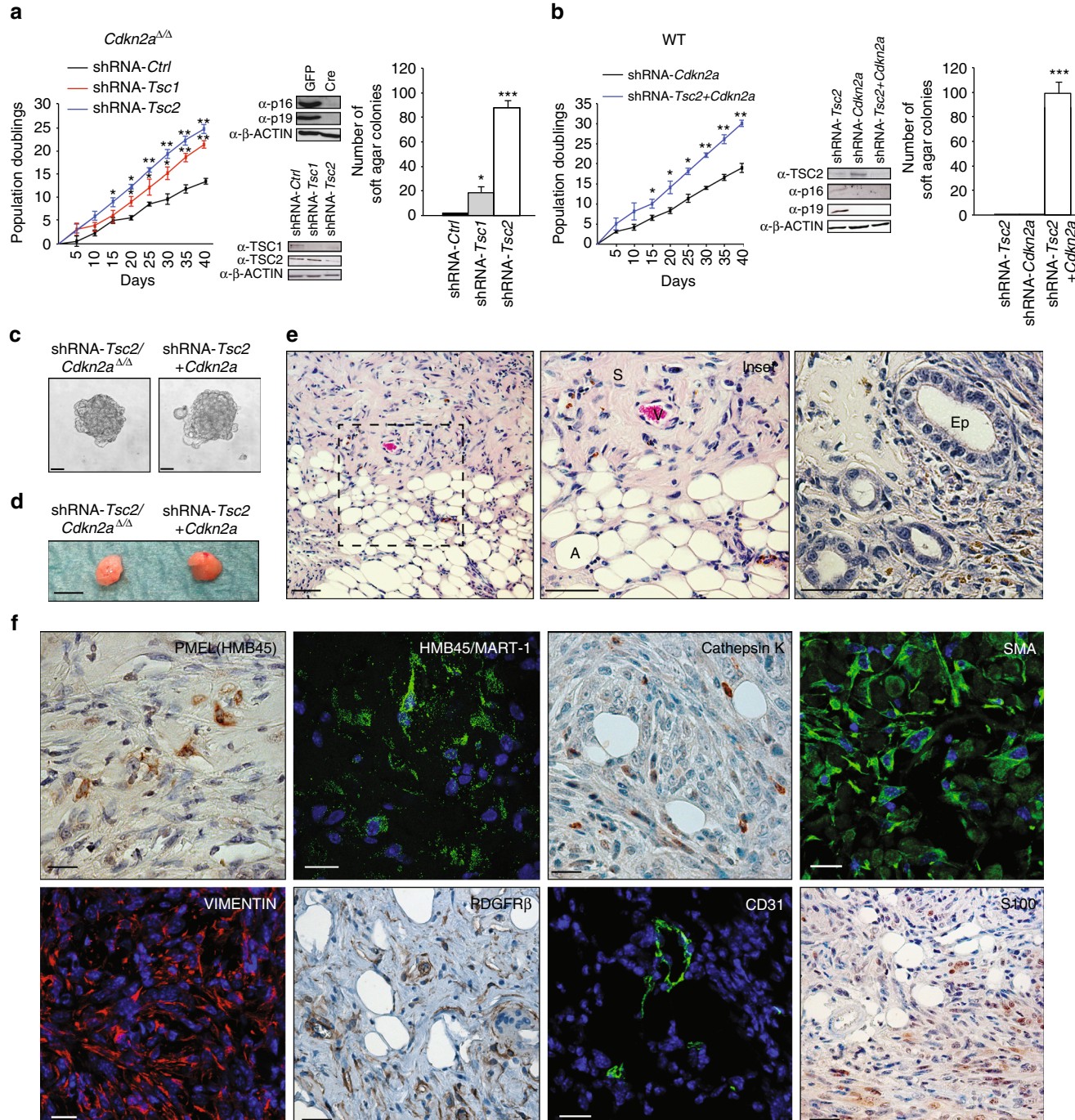

**Fig. 1** An ex vivo genetically engineered model of renal AML. **a** Proliferation assay, western blot and soft-agar assay of *Cdkn2a$^{\Delta/\Delta}$* primary kidney cells following infection with lentiviruses expressing an empty shRNA (shRNA-*Ctrl*) or shRNAs against *Tsc1* (shRNA-*Tsc1*) or *Tsc2* (shRNA-*Tsc2*). **b** Proliferation assay, western blot and soft-agar assay of primary kidney cells infected with lentiviruses expressing shRNAs against *Cdkn2a* (shRNA-*Cdkn2a*) or *Tsc2* and *Cdkn2a* (shRNA-*Tsc2* + *Cdkn2a*). All graphs depict mean ± s.d. Student's *t* test, $n = 3$. *$P < 0.05$, **$P < 0.01$, ***$P < 0.001$. **c** Phase contrast images of spheres formed in suspension culture conditions from shRNA-*Tsc2/Cdkn2a$^{\Delta/\Delta}$* and shRNA-*Tsc2* + *Cdkn2a* primary kidney cells. Scale bars: 50 μm. **d** Representative allografttumours derived from subcutaneous injections of respective sphere cells into *SCID-beige* mice.Scale bars: 1 cm. **e** Representative histology of allograft renal AML-like tumours. Characteristic histological features of AML are indicated by letters, where A shows adipocyte, S shows spindle cells and V shows an aberrant blood vessel. Epithelial structures are indicated by Ep. Scale bars: 50 μm. **f** Positive immunofluorescence or immunohistochemical staining in engineered renal AML-like allograft tumours for PMEL(HMB45), HMB45/MART-1, CATHEPSIN K, SMA, VIMENTIN, PDGFRβ, CD31 and S100. Scale bars: 50 μm

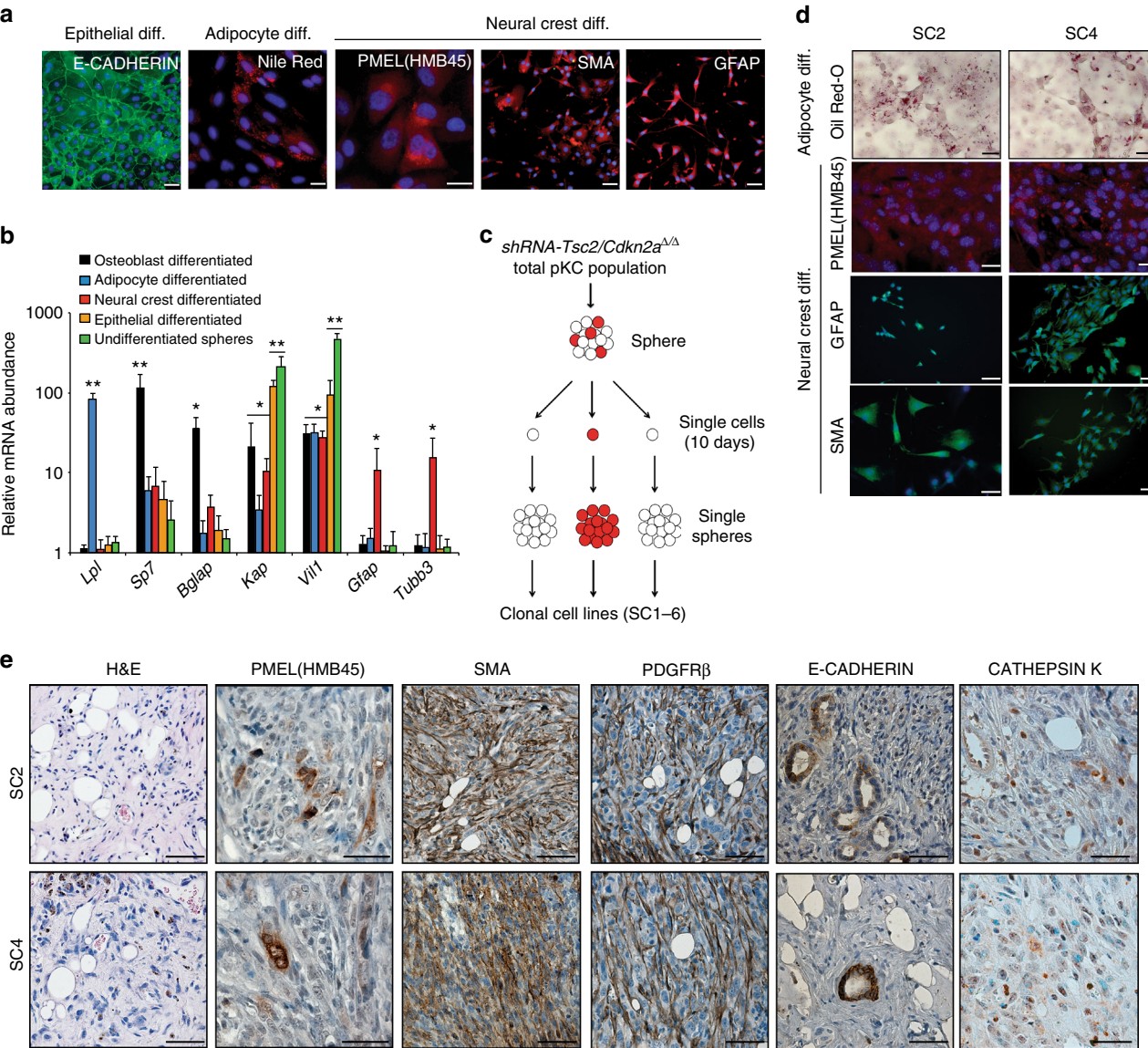

**Fig. 2** Cells from *Tsc2/Cdkn2a*deficient spheres exhibit stem cell-like characteristics. **a** shRNA-*Tsc2/Cdkn2a*$^{\Delta/\Delta}$ spheres were cultured in specific epithelial-, adipocyte- and neural crest-differentiation media and stained immunofluorescently using antibodies against E-CADHERIN, PMEL (HMB45), SMA or GFAP or stained with Nile Red. Scale bars: 50 µm. **b** mRNA expression analysis of the indicated genes in sphere cells differentiated using the indicated conditions compared to undifferentiated spheres and normalised to expression levels in freshly isolated primary kidney cells. Graph depicts mean ± s.d. Student's *t* test, *n* = 3. * *P* < 0.05, ** *P* < 0.01. **c** Experimental overview of the generation of clonal cell lines (Sphere Clone 1–6 (SC1–6)). **d** Two independent sphere clones (SC2 and SC4) were cultured in specific adipocyte- and neural crest-differentiation media and stained immunofluorescently using antibodies against PMEL (HMB45), GFAP or SMA or stained with Oil-Red-O. Scale bars: 50 µm.Analyses of two additional sphere clones (SC3 and SC6) are shown in Supplementary Fig. 7a. **e** Representative pictures of AML-like allograft tumours obtained from SC2 and SC4. Positive immunohistochemical stainings of PMEL(HMB45), SMA, PDGFRβ, E-CADHERIN and CATHEPSIN K. Scale bars: 50 µm. Analyses of tumours from two additional sphere clones are shown in Supplementary Fig. 7c

of mature adipose tissue, scattered blood vessels and spindle shaped cells (Fig. 1e). Many tumours additionally exhibited some epithelial tubular or cystic structures (Fig. 1e), although these made up only a very small component of the entire tumour. At the ultrastructural level, allograft tumours exhibited an expanded endoplasmic reticulum (Supplementary Fig. 3a–d) that is reminiscent of the expanded endoplasmic reticulum seen in human renal AML tumours[26]. Importantly, regions of adipocyte differentiation and spindle cells were immunoreactive for an antibody cocktail that recognises the melanocyte markers HMB45 and MART-1, for the melanocyte protein PMEL (the HMB45 antigen), SMA and VIMENTIN (Fig. 1f), reproducing the molecular characteristics that serve as diagnostic criteria for human renal AML. Approximately 15% of cells in our allograft tumours exhibited immunoreactivity for PMEL (Supplementary Fig. 3e). While the majority of cells in human renal AMLs typically stain strongly and diffusely for CATHEPSIN K[27], our allograft tumours exhibited a weak diffuse cytoplasmic positivity with ~10% of cells exhibiting strong immunoreactivity for CATHEPSIN K (Fig. 1f). mRNA analysis demonstrated high levels of expression of the melanocyte genes *Pmel*, *Tyrp1*, *Ctsk* (encoding CATHEPSIN K) and *Mlana* (also known as *Mart1*, encoding the MART-1 antigen) in allografts compared to normal kidney (Supplementary Fig. 3g) and western blotting confirmed that allograft tumours expressed both PMEL and CATHEPSIN K (Supplementary Fig. 3h, i). Similarly to human renal AMLs,

the allograft tumour cells were negative for the endothelial cell marker CD31, which instead labelled scattered dysmorphic blood vessels in the tumours (Fig. 1f). Human renal AMLs have been shown to harbour numerous cells that express PDGFRβ[10] and allograft tumours exhibited a similar staining pattern (Fig. 1f). Human renal AMLs stain positively for the neural stem cell marker NG2[28] and our allograft tumours similarly exhibited diffuse NG2 immunoreactivty (Supplementary Fig. 4a). While mature normal adipocytes stain positively for S100, adipocytes in human renal AMLs typically do not express S100, but other non-adipocyte components of the tumour have been reported to stain positively in about 28% of renal AML cases[29]. Consistent with this, in our allograft tumours, adipocytes stained negatively for S100 and spindle shaped cells stained positively (Fig. 1f). In summary, these ex vivo genetically engineered cells derived from a kidney-resident cell type represent an allograft model that shares many histological and molecular features with human renal AML or AMLEC.

We next sought to determine whether the AML-like tumour and cellular phenotypes observed in our system are specifically related to the *Tsc2* and *Cdkn2a* loss-of-function genotype or whether any genetic combination that causes transformation and sphere formation would give similar results. In the context of another research project that will be described in detail elsewhere, we discovered that the overexpression of the *Myc* oncogene in Adeno-Cre-treated renal cell cultures derived from *Trp53$^{fl/fl}$* mice (*Myc/Trp53$^{\Delta/\Delta}$*) causes the formation of spheres and these cells form tumours in allograft assays (Supplementary Fig. 4c–f). In contrast to the *Tsc2/Cdkn2a* AML-like tumours, *Myc/Trp53$^{\Delta/\Delta}$* allograft tumours grew as solid masses of epithelial-like cells, sometimes organised in a pseudo-acinar pattern. Tumour cells typically had low cytoplasmic volumes with enlarged prominent nuclei. No histological similarities with renal AML were evident and tumours showed no immunoreactivity for PMEL (Supplementary Fig. 3e, f), S100 or NG2 (Supplementary Fig. 4a, b). These findings further support the specificity of the effect of *Tsc2* loss-of-function in our engineered renal AML model system.

**AML-forming cells have neoplastic stem cell properties.** Given that AML-like tumours derived from our engineered cells comprise multiple different types of cells, as well as the previous proposal that AMLs arise from a neural crest-derived cell type[5], we cultured shRNA-*Tsc2/Cdkn2a$^{\Delta/\Delta}$* sphere cells in specific adipocyte-, osteoblast-, neural crest- and epithelial- differentiation media to assess their capacity to form different cellular lineages in culture. Cells from spheres could be differentiated into cells that expressed markers of adipocytes (Nile Red and Oil-red-O staining), glia (GFAP), neurons (TUBULIN-β3), melanocytes (HMB45/MART-1), smooth muscle cells (SMA) and epithelial cells (E-CADHERIN) (Fig. 2a). The observation of a melanocyte-like phenotype was supported by the fact that shRNA-*Tsc2* + *Cdkn2a* sphere cells that were differentiated in neural-crest differentiation medium showed similar positive staining for CATHEPSIN K, PMEL, MITF and MART-1 to the B16F1 melanoma cell line (Supplementary Fig. 5). Our findings based on stainings were supported by mRNA expression analyses (Fig. 2b) showing that adipocyte differentiation medium induced expression of the adipocyte-specific gene *Lpl*, osteoblast differentiation medium induced expression of the osteoblast-specific genes *Sp7* and *Bglap* and neural crest differentiation medium induced the expression of the glial-specific gene *Gfap* and neuron-specific gene *Tubb3*. Undifferentiated spheres exhibit high levels of expression of epithelial-specific genes (see later results) including *Kap* and *Vil1*, which remained elevated when cultured in epithelial differentiation medium but were greatly reduced when

cultured in the other differentiation media. To further demonstrate the specificity of this phenotype we differentiated shRNA-*Tsc2* + *Cdkn2a* spheres in parallel with cultures of shRNA-*Cdkn2a* kidney cells (which do not form spheres) and cultures from *Myc/Trp53$^{\Delta/\Delta}$* sphere cells. Only the shRNA-*Tsc2* + *Cdkn2a* spheres but not the two negative control cell lines could be differentiated into Nile Red or PMEL positive cells (Supplementary Fig. 6a) nor did the differentiated cultures upregulate any of the adipocyte, osteoblast or neural-crest lineage specific genes that are induced upon differentiation of shRNA-*Tsc2* + *Cdkn2a* spheres (Supplementary Fig. 6b). The two negative control cell lines also showed no expression of PMEL by western blotting (Supplementary Fig. 6c).

To determine whether single cells within spheres have the stem cell characteristic of being able to differentiate into these multiple cellular lineages we disrupted cells from individual shRNA-*Tsc2/Cdkn2a$^{\Delta/\Delta}$* spheres and plated them as single cells. These single cells generated new spheres within 10 days and these were passaged to form clonal cell lines (termed SC1- SC6) (Fig. 2c). Four of six independent single cell-derived clonal lines derived from three independent starting spheres possessed the ability to differentiate into osteoblasts, adipocytes, glia and melanocytes in culture, assessed using stainings (Fig. 2d and Supplementary Fig. 7a) and mRNA expression (Supplementary Fig. 7b). These four clones also formed characteristic AML tumours in in vivo allograft assays (six tumours from six injections of each clone) (Fig. 2e and Supplementary Fig. 7c), showing regions of adipocyte differentiation, spindle cells, blood vessels and epithelial structures. These tumours showed similar patterns of immunoreactivity to tumours obtained from whole sphere populations when stained with antibodies against PMEL, SMA, PDGFRβ, CATHEPSIN K, NG2 and S100, as well as displayed E-CADHERIN positive epithelial structures (Fig. 2e and Supplementary Figs. 4a, b and 7c). The frequency of PMEL-positive cells in allograft tumours was similar to the frequency seen in allograft tumours from whole populations of spheres (Supplementary Fig. 3f). Allograft tumours from SC2 and SC4 express melanocyte genes (Supplementary Fig. 3g) and the proteins PMEL, MiTF and CATHEPSIN K (Supplementary Fig. 7d). These results demonstrate that single cells from our reverse-engineered sphere cell populations have properties of tumour-forming AML stem cells that have the ability to differentiate into several neural crest-derived and mesenchymal cellular lineages, as well as into epithelial cells. Interestingly, two of the six clones (SC1, SC5) did not form tumours in allograft experiments (zero tumours from six injections of each clone). These cells could be differentiated into Nile Red-positive and SMA-positive cells in appropriate media, but could not be differentiated into PMEL- or GFAP-expressing cells (Supplementary Fig. 7e), demonstrating that not all cells within a sphere have the properties of AML neoplastic stem cells, potentially implying the existence of a hierarchy of differentiation.

Given this mesenchymal differentiation phenotype we next analysed markers of mesenchymal stem cells in our engineered sphere cells. Flow cytometric comparisons of sphere-selected shRNA-*Tsc2/Cdkn2a$^{\Delta/\Delta}$* cells and control *Cdkn2a$^{\Delta/\Delta}$* kidney cells (Supplementary Fig. 8a) revealed that both genotypes of sphere cells expressed SCA-1 and CD29, and shRNA-*Tsc2/Cdkn2a$^{\Delta/\Delta}$* cells specifically expressed the general mesenchymal marker CD73, but none of the cells expressed other characteristic markers of mesenchymal stem cells (CD105, CD106). These findings argue against the idea that we have simply isolated a normal population of kidney-resident mesenchymal stem cells that represent the AML-forming sphere cells in our culture system. The absence of expression of *Nanog*, *Sox2* and *Oct4*, which are abundantly expressed in ES cells (Supplementary

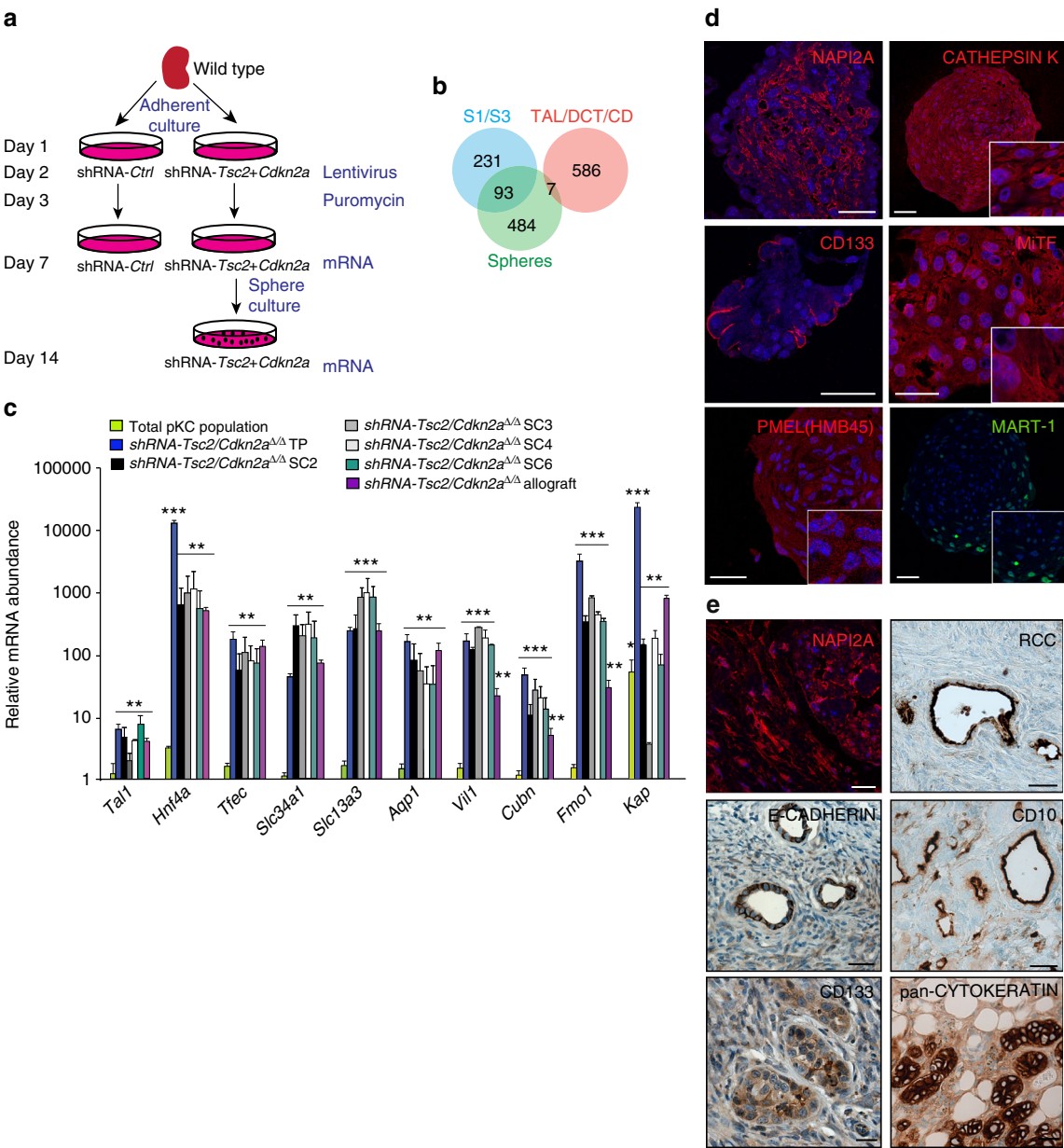

**Fig. 3** Engineered sphere cells exhibit gene expression patterns of proximal tubule epithelial cells. **a** Overview of experimental procedure to identify the gene expression profile of engineered shRNA-Tsc2 + Cdkn2a AML-forming sphere cells. **b** Venn diagram showing the overlap of gene expression signature of engineered spheres (green) with distinct mRNA signatures of S1 and S3 segments of proximal tubules (blue) or a pooled mRNA signature of DCT (distal convoluted tubule), TAL (thick ascending limb of the Loop of Henle) and CD (collecting duct) (red). **c** Real time PCR analysis of the indicated genes in cultures of control kidney cells (total pKC population), total sphere populations, sphere cultures of SC2, SC3, SC4 and SC6 and AML-like allograft tumours. Graph depicts mean ± s.d. Student's t test, n = 3. **P < 0.01; ***P < 0.001. **d** Positive immunofluorescence stainings of spheres using antibodies against NAPI2A, CATHEPSIN K, pan-CD133, MiTF, PMEL(HMB45) and MART-1. **e** Positive immunofluorescence and immunohistochemical stainings of AML-like allograft tumours using antibodies against NAPI2A, RCC, E-CADHERIN, CD10, CD133 and pan-CYTOKERATIN. Scale bars: 50 μm

Fig. 8b), also argues against an induced pluripotent stem cell phenotype of our engineered sphere cells.

**Neoplastic AML stem cells derive from renal epithelial cells**. To gain further insight into molecular features of the engineered AML-initiating cells we used RNA sequencing to compare gene expression between primary renal cell cultures 5 days after infection with control or shRNA-Tsc2 + Cdkn2a viruses and spheres derived from shRNA-Tsc2 + Cdkn2a-infected cells harvested 12 days after infection (Fig. 3a). Gene expression values are provided in Supplementary Data 1. Compared with control infected cells, total populations of shRNA-Tsc2 + Cdkn2a cells

displayed 154 genes that were upregulated >8-fold. Gene set enrichment analyses (GSEA) revealed association with expression signatures related to cellular proliferation and mTOR signalling (Supplementary Data 2), consistent with the introduced knockdowns and elevated proliferation rate of these cultures. Comparison of shRNA-Tsc2 + Cdkn2a spheres to shRNA-Tsc2 + Cdkn2a total cell populations (similar results were obtained when comparing shRNA-Tsc2 + Cdkn2a spheres to control infected cells, data not shown) identified 584 genes that were >8-fold upregulated in spheres (Supplementary Data 1). Interestingly, 93 of these genes overlapped with a set of 324 genes that are expressed at 10-fold or higher levels in S1 or S3 segments of

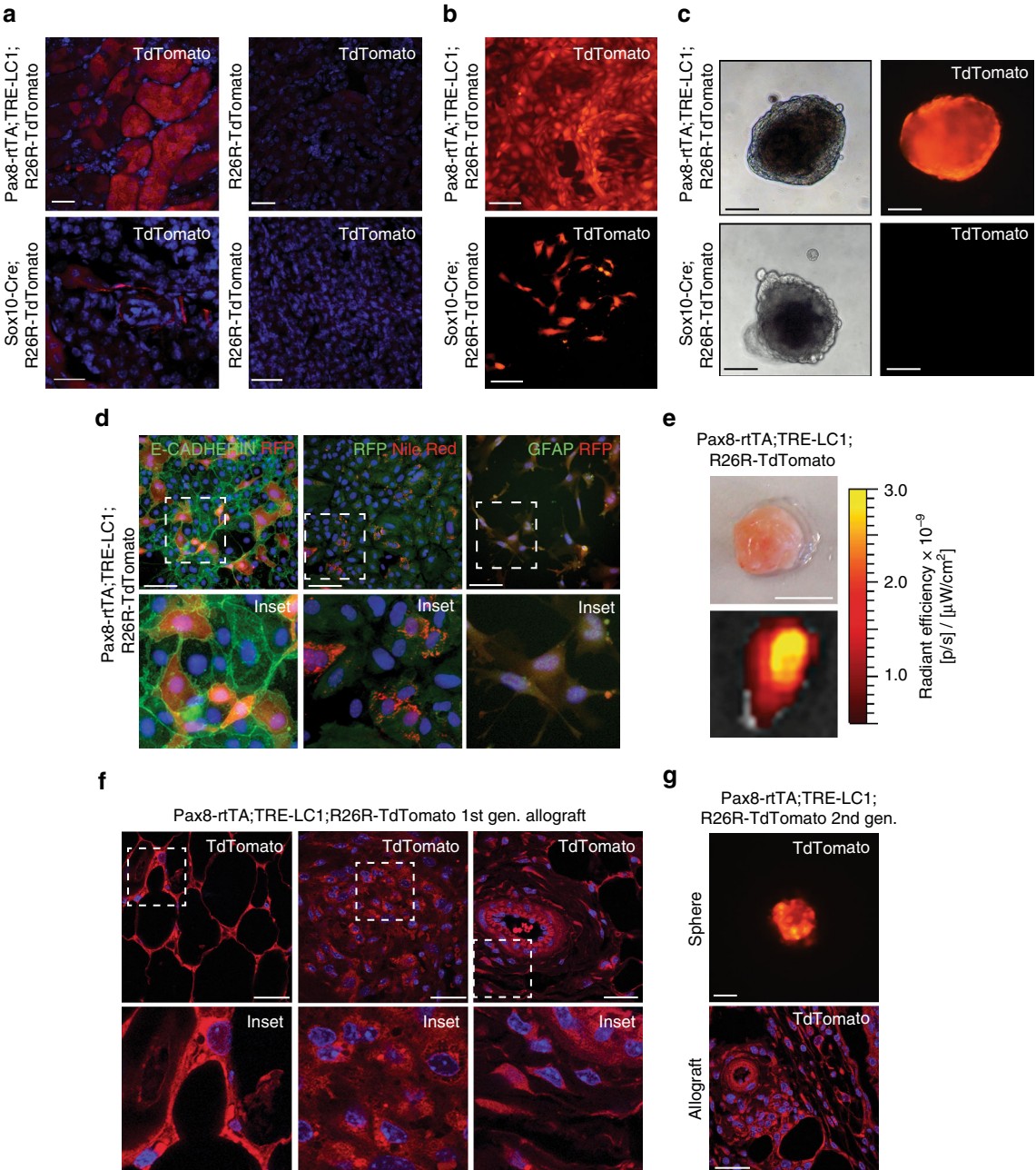

**Fig. 4** Engineered renal AML-like tumours derive from renal epithelial cells. **a** Fluorescence images of kidney tissue from doxycycline-treated *Pax8-rtTA;TRE-LC1;R26R-td Tomato* mice and from *Sox10-Cre;R26R-td Tomato* mice and respective controls. Scale bars: 50 μm. **b** Images of primary kidney cells isolated from doxycycline-treated *Pax8-rtTA;TRE-LC1;R26R-td Tomato* mice and from *Sox10-Cre R26R-td Tomato* mice. Scale bars: 50 μm. **c** Red fluorescence signals in spheres derived from infection of cultures in **b** with shRNA-*Tsc2* + *Cdkn2a* lentivirus. Scale bars: 50 μm. **d** Differentiation of Pax8-td Tomato positive spheres into epithelial cells (E-CADHERIN immunofluorescence), adipocytes (Nile Red staining) and glial cells (GFAP immunofluorescence). tdTomato was detected using an anti-RFP antibody. Scale bars: 50 μm. **e** tdTomato-positive shRNA-*Tsc2* + *Cdkn2a* allograft derived from *Pax8-rtTA;TRE-LC1;R26R-td Tomato* spheres. Scale bars: 1 cm. **f** Representative red fluorescence in histological sections of shRNA-*Tsc2* + *Cdkn2a* allografts derived from *Pax8-rtTA;TRE-LC1;R26R-td Tomato* mice: DAPI and td Tomato overlay. Scale bars: 50 μm. Lower panels represent zooms of the boxed regions in the upper panels. **g** Example of second-generation shRNA-*Tsc2* + *Cdkn2a* td Tomato positive sphere and allograft histological section derived from a primary shRNA-*Tsc2* + *Cdkn2a* allograft that originated from a *Pax8-rtTA;TRE-LC1;R26R-td Tomato* mouse. Scale bars: 50 μm

proximal tubules compared with distal tubules, loops of Henle and collecting ducts[30] (Fig. 3b). In contrast, comparison with a cumulated list of 593 genes that are specifically upregulated either in distal tubules, loops of Henleor collecting ducts compared to proximal tubules[30] revealed an overlap of only 7 genes (Fig. 3b). The upregulation of a subset of the identified proximal tubule-specific genes (*Slc34a1*, *Slc13a3*, *Aqp1*, *Vil1*, *Cubn*, *Fmo1*, *Kap*) in

spheres compared with total shRNA-*Tsc2*/sh+*Cdkn2a* populations was confirmed by real time PCR analyses (Fig. 3c). This result surprisingly demonstrates that spheres exhibit a stronger renal proximal tubule epithelial gene expression signature than cultured primary renal epithelial cells. Further supporting the epithelial nature of the spheres, *Hnf4a* was upregulated ~10,000 fold in spheres (Fig. 3c). *Hnf4a* encodes a transcription factor that

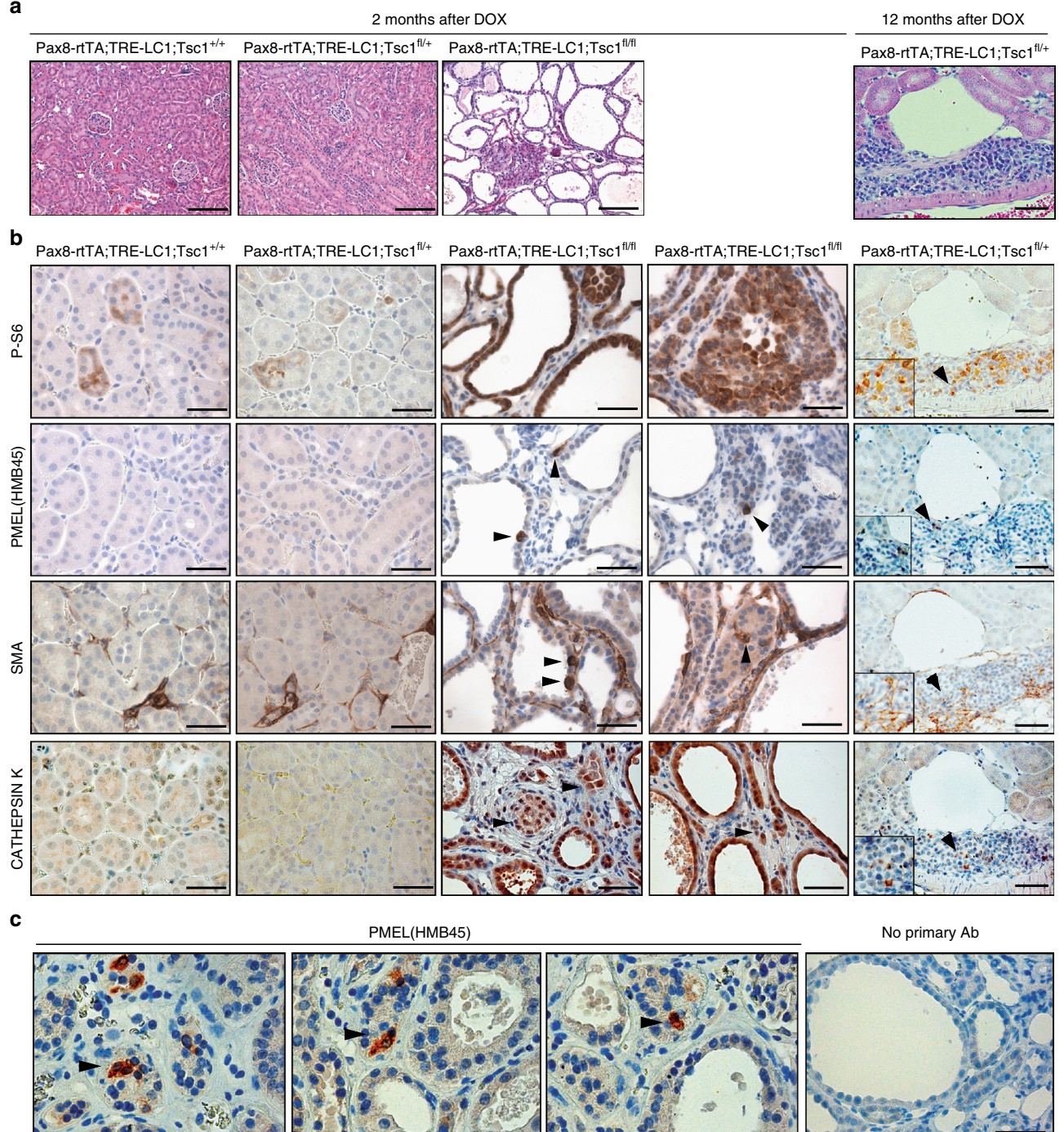

**Fig. 5** Renal epithelium-specific deletion of *Tsc1* causes AML-like differentiation in mouse kidneys. **a** Representative histological images of kidneys 2 months after doxycycline treatment of *Pax8-rtTA;TRE-LC1;Tsc1*[+/+] (wild type), *Pax8-rtTA;TRE-LC1;Tsc1*[fl/+] (heterozygous *Tsc1* mutant), *Pax8-rtTA;TRE-LC1;Tsc1*[fl/fl] (homozygous *Tsc1* mutant) mice and in the fifth panel, images of kidneys 12 months after doxycycline treatment of *Pax8-rtTA;TRE-LC1;Tsc1*[fl/+]. Scale bars: 150 μm. **b** Representative immunohistochemical stainings of kidney sections from mice of the indicated genotypes using antibodies against phosphorylated Ser240/244 ribosomal protein S6 (P-S6), PMEL (HMB45), SMA and CATHEPSIN K. Arrowheads in the third, fourth and fifth columns highlight positive cells lining cysts or in regions of adenoma-like proliferation. Scale bars: 50 μm. **c** High magnification pictures of positive cytoplasmic immunohistochemical staining of PMEL (HMB45) in *Pax8-rtTA;TRE-LC1;Tsc1*[fl/fl] kidneys 2 months after doxycycline treatment. Scale bars: 50 μm

is necessary for development of renal proximal tubules, is highly expressed in proximal tubule epithelial cells and regulates the expression of numerous proximal tubule specific genes[31,32]. GSEA analyses also identified experimentally established HNF4A and HNF1A target genes (Supplementary Data 2) and genes with bioinformatically predicted HNF4A and HNF1A transcription

factor binding sites (Supplementary Data 2) as being significantly enriched in the sphere gene set. Genes encoding 14 different transcription factors that have been implicated in the development of many different cellular lineages were also upregulated >8-fold in spheres (Supplementary Data 1) and 3 of these (*Hnf4a, Tal1, Tfec*) were confirmed by real time PCR analyses (Fig. 3c).

These proximal tubule and transcription factor gene signatures were also present in the four clonal sphere lines and in AML allograft tumours (Fig. 3c), suggesting that the engineered tumours display at least some of the characteristic gene expression profile of the proximal tubule epithelial cell, arguing that proximal tubular epithelial cells are the origin of these engineered tumours. Further supporting this notion, some cells in spheres expressed NAPI2A or stained positively for pan-CD133 (not the glycosylation-specific 'stem cell marker' antibody) (Fig. 3d), which are expressed by normal renal proximal tubule epithelial cells. At the same time, shRNA-$Tsc2$ + $Cdkn2a$ spheres also exhibited evidence of an AML or melanocyte-like phenotype as shown by immunoreactivity for the AML markers PMEL, CATHEPSIN K, MiTF and MART-1 (Fig. 3d), expression of MiTF and CATHEPSIN K by western blotting (Supplementary Fig. 9a) and upregulation of $Tyrp1$ gene expression in comparison to non-sphere populations or to $Myc/Trp53^{\Delta/\Delta}$ sphere populations (Supplementary Fig. 9b). Spindle cells in allograft AML tumours (Fig. 3e) also expressed the proximal tubule marker protein NAPI2A and epithelial structures in these tumours were positive for the epithelial markers E-CADHERIN and PAN-CYTOKERATIN and stained positively for pan-CD133, RCC and CD10 (Fig. 3e), three markers of the renal proximal tubule epithelium. We conclude that AML-forming sphere cells can also differentiate into proximal tubule epithelial cells. Consistent with a renal epithelial phenotype, scattered cells in allograft tumours derived from total sphere populations or from sphere clonal cell lines expressed the transcription factor PAX8, which is normally expressed in epithelia of the renal and genital-urinary tract (Supplementary Fig. 9c).

To gain further evidence for an epithelial origin of our engineered tumours we next genetically marked renal epithelial cells in vivo, labelling all nephron segments by doxycycline feeding of mice harbouring transgenes encoding $Pax8$ promoter-regulated expression of rtTA, Tet-responsive Cre expression[33] and ROSA26-lox-STOP-lox-regulated tdTomato expression[34] ($Pax8$-$rtTA;TRE$-$LC1;R26R$-$LSL$-$tdTomato$) (Fig. 4a). In total 85% of cells cultured from these kidneys were tdTomato-positive (Fig. 4b) and when transformed with shRNA-$Tsc2$ + $Cdkn2a$ lentiviruses 90% of the resulting spheres were tdTomato-positive (Fig. 4c). These could be differentiated to form tdTomato-positive epithelial cells, adipocytes and glia (Fig. 4d) and expressed characteristic genes of differentiated lineages (Supplementary Fig. 6b). Negative staining controls are shown in Supplementary Fig. 10a–d. Allograft tumours derived from these cultures expressed tdTomato (Fig. 4e) and cells in tumour regions characterised by adipocyte, spindle cell and thick-walled blood vessel histology were labelled with tdTomato (Fig. 4f). Similarly to the other tumours generated in this study, these allograft tumours stained positively for PMEL, CATHEPSIN K, PDGFRβ, SMA and VIMENTIN (Supplementary Fig. 10e), expressed PMEL by western blotting (Supplementary Fig. 10f), and exhibited a similar frequency of PMEL expressing cells (Supplementary Fig. 10g).

Cells isolated by collagenase digestion of resected tumours could be cultured as tdTomato-expressing spheres and sphere-selected cells formed new tdTomato-positive allograft AML tumours (Fig. 4g), establishing that these cells have tumour-propagating capacity, an expected feature of a neoplastic tumour stem cell. In contrast, in vivo lineage tracing of neural crest-derived cells using $Sox10$-$Cre;R26R$-$LSL$-$tdTomato$ mice labelled a set of non-tubular interstitial cells in the intact kidney (Fig. 4a) that were present as rare clusters of tdTomato-labelled cells in our kidney cultures (Fig. 4b). However, after infection with shRNA-$Tsc2$ + $Cdkn2a$ lentiviruses none of these cells contributed to spheres (Fig. 4c) or tumours (Supplementary Fig. 10h), excluding

that they are the cells of origin of our AML model. These studies collectively demonstrate that loss of function of $Tsc2$ and $Cdkn2a$ in cultured proximal tubule epithelial cells converts them into an AML-forming neoplastic stem cell.

Guided by these results, we repeated a previously published experiment involving $Pax8$-$Cre$-driven inducible deletion of $Tsc1$ in adult renal epithelia using $Pax8$-$rtTA;TRE$-$LC1;Tsc1^{fl/fl}$ mice[33]. As reported, within 2 months of homozygous deletion of $Tsc1$ in 6–8 week old mice, kidneys developed a severe polycystic phenotype with intermixed regions of adenoma-like epithelial proliferation but did not develop renal AMLs (Fig. 5a). All cystic and adenoma-like regions stained strongly for phosphorylation of ribosomal protein S6 (Ser240/244), indicative of $Tsc1$ deletion (Fig. 5b). Interestingly, however, immunohistochemical analyses revealed the presence of scattered PMEL positive cells that were present as single cells or clusters of 2–3 cells (on average 35 sites per mouse kidney section) within regions of solid or cystic epithelial proliferation (Fig. 5b). PMEL staining was predominantly cytoplasmic (Fig. 5c). PMEL positive cells were never observed in kidneys from wild type mice or $Tsc1$ heterozygous mutant mice ($Pax8$-$rtTA;TRE$-$LC1;Tsc1^{fl/+}$). Cysts and adenoma lesions also showed scattered cells expressing SMA, another marker of renal AMLs (Fig. 5b). CATHEPSIN K staining was upregulated in all cells in cysts and adenomas (Fig. 5b), and CATHEPSIN K protein expression was detected by western blotting in kidney lysates from homozygous, but not heterozygous $Tsc1$ mutant mice or wild type mice (Supplementary Fig. 3i), suggesting that this increase is directly related to $Tsc1$ deletion and consistent with the fact that almost all cells in human renal AMLs express CATHEPSIN K[27]. Given that the strong hyperproliferative phenotype of $Tsc1$ homozygous deletion prevented aging experiments, we generated a cohort of 12 $Pax8$-$rtTA;TRE$-$LC1;Tsc1^{fl/+}$ heterozygous mutant mice and analysed them 1 year after inducing gene deletion in 6–8 week old mice. By histological sectioning of kidneys we identified 20 cysts (14 were lined by a single layer of epithelial cells, 6 cysts displayed atypical, multilayered morphology), 5 small tumours with epithelial morphology and 28 lesions comprising closely packed cells with low cytoplasmic volume, often occurring close to blood vessels (Supplementary Fig. 11). These lesions exhibit a similar morphology to lesions that were identified upon $Tsc1$ deletion under a ubiquitious promoter and that were proposed to represent potential precursors of renal AMLs[22]. Consistent with this idea, the lesions in our mice harboured many cells that were positive for phosphorylation of ribosomal protein S6 (Ser240/244) and some cells that were positive for CATHEPSIN K, SMA, and more rarely for PMEL (Fig. 5b). Thus, the epithelial-specific PAX8-Cre driver appears to recreate a similar phenotype to that previously described with the ubiquitous Cre driver. These observations provide in vivo evidence to show that loss of TSC function in renal epithelial cells can cause the formation of different lesions that contain cells that express several different molecular markers of renal AML, consistent with our ex vivo cell engineering studies.

**Epithelial origin of human AMLs.** We next sought to determine the relevance of these mouse-based findings to human renal AML. To overcome the senescence barrier associated with loss of TSC function in primary human cells, we employed RPTECs immortalised with SV40 large T Antigen and hTERT[35] and infected these cells with four independent shRNA-$TSC2$ lentiviruses, efficiently reducing TSC2 protein abundance (Supplementary Fig. 12a) and inducing the formation of spheres in non-adherent culture conditions (Supplementary Fig. 12b). These spheres could be differentiated into epithelial cells, adipocytes and

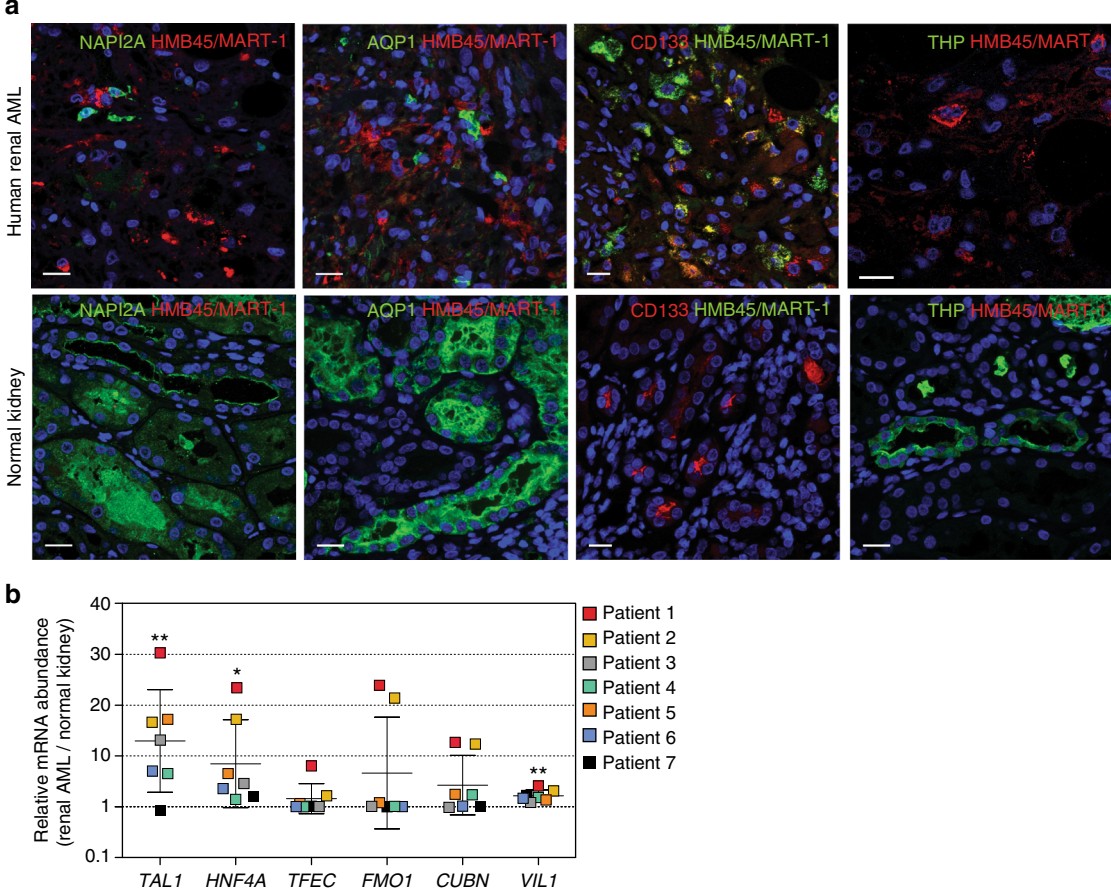

**Fig. 6** Human AML tumours express proximal tubule markers. **a** Representative immunofluorescence co-stainings for the melanocytic/AML marker HMB45/MART-1 and the proximal tubule epithelial markers NAPI2A, AQP1 and CD133 and distal tubule epithelial marker Tamm Horsfall glycoprotein (THP) in human renal AMLs (top row) and normal kidney (bottom row). Scale bars: 50 μm. **b** Relative mRNA abundance (renal AML vs. matched normal kidney) of three selected transcription factors (*TAL1*, *HNF4A* and *TFEC*) and three proximal tubule markers (*FMO1*, *CUBN* and *VIL1*) in seven human AML cases. The graph shows the values for each patient and the whisker plot depicts mean ± s.d. * denotes $P < 0.05$ and **$P < 0.01$ (Student's *t* test) statistically significant different expression between AML and normal

glia in culture (Supplementary Fig. 12c). Importantly, while immortalised RPTECs did not form sub-cutaneous tumours (*n* = 3 mice), two independent TSC2 hairpins (II and IV) caused the formation of xenograft tumours (*n* = 4 mice for each hairpin) that histologically resembled the characteristic spindle cell, vascular and adipocyte structures of human renal AMLs, and also displayed regions of epithelial structures (Supplementary Fig. 12d). These tumours expressed the AML marker proteins HMB45, SMA, VIMENTIN, PDGFRβ and epithelial structures were positive for E-CADHERIN (Supplementary Fig. 12d). In comparison to control cells, sh*TSC2* spheres exhibited increased mRNA expression of proximal tubule signature genes and transcription factor encoding genes that were upregulated in spheres engineered from mouse cells (Supplementary Fig. 12e). Thus, cultured human renal proximal tubule epithelial cells are also converted to AML-forming cells by loss of function of TSC2 in cooperation with abrogation of cellular senescence.

Immunofluorescence staining of seven sporadic human AML tumours revealed that all tumours contained cells that express proteins that are expressed in proximal tubular epithelial cells in the kidney, namely NAPI2A, pan-CD133 and AQP1, but did not contain cells expressing THP, a marker of epithelial cells of the thick ascending limb of the loop of Henle (Fig. 6a). Analyses of mRNA isolated from matched pairs of samples of normal kidney and AML tumour (*n* = 7 cases) from paraffin-embedded tissue

revealed that all AMLs expressed similar or higher levels of proximal tubule marker genes (*FMO1*, *CUBN*, *VIL1*) and several transcription factor-encoding genes identified from our mouse studies (*TAL1*, *HNF4A*, *TFEC*) as normal kidney (Fig. 6b). These findings corroborate our mouse tumour engineering findings and are consistent with the idea that human renal AMLs derive from renal epithelial cells.

We next obtained a set of isogenic renal AML cell lines to investigate whether these cells display similarities to our engineered models. TRI-102 cells are an immortalised (E6/E7 and hTERT) derivative[36] of a *TSC2* null primary cell culture that was derived from a human renal AML[37]. TRI-103 is a derivative of TRI-102 into which TSC2 expression has been re-introduced[36]. Similarly to our engineered cells, TRI-102 but not TRI-103 cells formed spheres when grown in epithelial medium on non-adherent plates (Fig. 7a). When grown as adherent cultures, both TRI-102 and TRI-103 expressed the mesenchymal marker VIMENTIN, as well as the epithelial marker E-CADHERIN (Fig. 7b). TRI-102 cells also expressed the proximal tubule marker proteins CD133 and NAPI2A and the expression of these proteins was absent or reduced in TRI-103 cells (Fig. 7b). TRI-102 cells when grown as adherent cells or as spheres expressed higher mRNA abundance than TRI-103 of the transcription factors and some of the proximal tubule genes that we identified in our engineered systems (Fig. 7c). Finally, TRI-102 but not TRI-

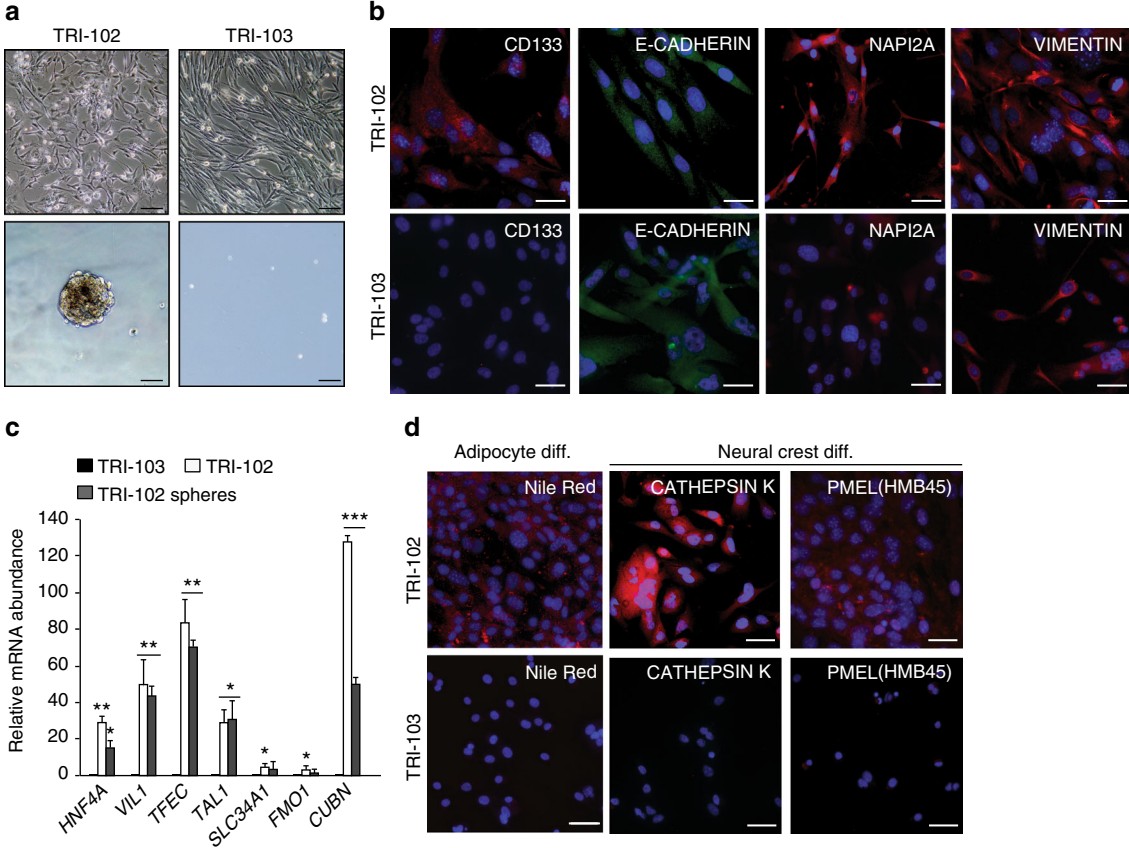

**Fig. 7** Human TSC-associated renal AML cells exhibit renal proximal tubule epithelial features. **a** Phase contrast images of TRI-102 and TRI-103 human TSC-associated renal AML cells growing on adherent plastic (upper panel) and spheres formed in suspension culture (lower panel). Scale bars 50 µm. **b** Immunofluorescence stainings for CD133, E-CADHERIN, NAPI2A and VIMENTIN in adherent TRI-102 (top row) and TRI-103 cells (bottom row). Scale bars: 50 µm. **c** Relative mRNA abundance of the indicated genes in adherent TRI-103 and TRI-102 cells and TRI-102 spheres. Graph depicts mean ± s.d. Student's $t$ test, $n = 3$. **$P < 0.01$; ***$P < 0.001$. **d** Differentiation of TRI-102 and TRI-103 cells into adipocytes (Nile Red staining) and neural crest derived cells (CATHEPSIN K and PMEL (HMB45) immunofluorescence). Scale bars: 50 µm

103 cells could be differentiated into Nile Red, CATHEPSIN K and PMEL positive cells in appropriate media (Fig. 7d). These findings establish that a cell line derived from a human renal AML have similar renal proximal tubule epithelial features and differentiation capacities to our engineered cellular systems.

**Therapeutic studies using engineered AML models.** In a recent phase 3 clinical trial of the mTORC1 inhibitor everolimus for large renal AMLs (>3 cm diameter) in TSC patients, 42% of the patients displayed a reduction in tumour volume of >50% within one year of treatment[20] and this therapy is now a recommended treatment for these patients. However, once everolimus therapy is discontinued, AML lesions typically resume growth[38,39]. The cellular consequences of everolimus therapy in TSC patient renal AML tumours have not been able to be studied. To attempt to address this issue we utilised our engineered AML models. Everolimus caused a greater inhibition of proliferation of shRNA-$Tsc2/Cdkn2a^{\Delta/\Delta}$ sphere-derived cells compared to control $Cdkn2a^{\Delta/\Delta}$ kidney cells when grown under adherent cell culture conditions (Fig. 8a). Everolimus also decreased the number and size of spheres in non-adherent culture conditions (Fig. 8b–d) without decreasing cellular viability (Fig. 8e). Rather, everolimus decreased the proportion of S-phase cells and caused accumulation of cells in the G1 and G2/M cell cycle phases (Fig. 8f). The addition of everolimus to cellular differentiation experiments did not alter the ability of shRNA-$Tsc2/Cdkn2a^{\Delta/\Delta}$ sphere cells to differentiate into different cellular lineages (Fig. 8g). Everolimus

treatment (10 mg/kg i.p.daily for 2 weeks) of mice harbouring shRNA-$Tsc2/Cdkn2a^{\Delta/\Delta}$ allograft tumours blocked tumour growth and tumours regressed after cessation of treatment but regrew after ~3 months (Fig. 8h). This behaviour mimics the clinical response to mTOR inhibitor therapy. Tumours harvested during everolimus treatment (Fig. 8i) exhibited no apparent differences in histological appearance compared with untreated tumours (Fig. 8j), nor in the abundance of PMEL immunoreactive cells (Fig. 8j, k) but contained fewer proliferating cells as assessed by BrdU labelling (Fig. 8l, m). While everolimus caused a 3-fold increase in the number of cleaved caspase-3 positive cells, a marker of apoptosis (Fig. 8n), it should be noted that the frequency of apoptotic cells was very low (<0.1% of cells) and is therefore unlikely to be the major contributor to the therapeutic effects of the drug. This observation is consistent with the cell culture data and demonstrates that mTORC1 inhibition predominantly induces cell cycle arrest of AML tumour cells, without any obvious effects on cellular differentiation, consistent with the observed absence of complete cures in renal AML patients.

**Discussion**

Biallelic loss-of-function mutations in $TSC2$ and more rarely in $TSC1$ appear to represent the key genetic driving events of renal AMLs[19]. We show that loss of function of TSC2 in genetic backgrounds that abrogate senescence and proliferation arrest is sufficient to convert mouse and human renal proximal tubular epithelial cells into neoplastic stem cells that give rise to allograft

or xenograft tumours that reproduce many of the histological and molecular features of human renal AMLs, providing the first models of AML-like tumours. It should be noted that these tumours represent 'models' as the genetic backgrounds that we employed in this study to overcome cellular senescence induced

by loss of TSC2 function are apparently not reflected in the genetics of human renal AMLs, for example, genetic mutations or deletions of *TP53*, *CDKN2A* or other senescence regulating tumour suppressor genes have not been observed in AML tumours. It remains possible that yet-to-be-identified

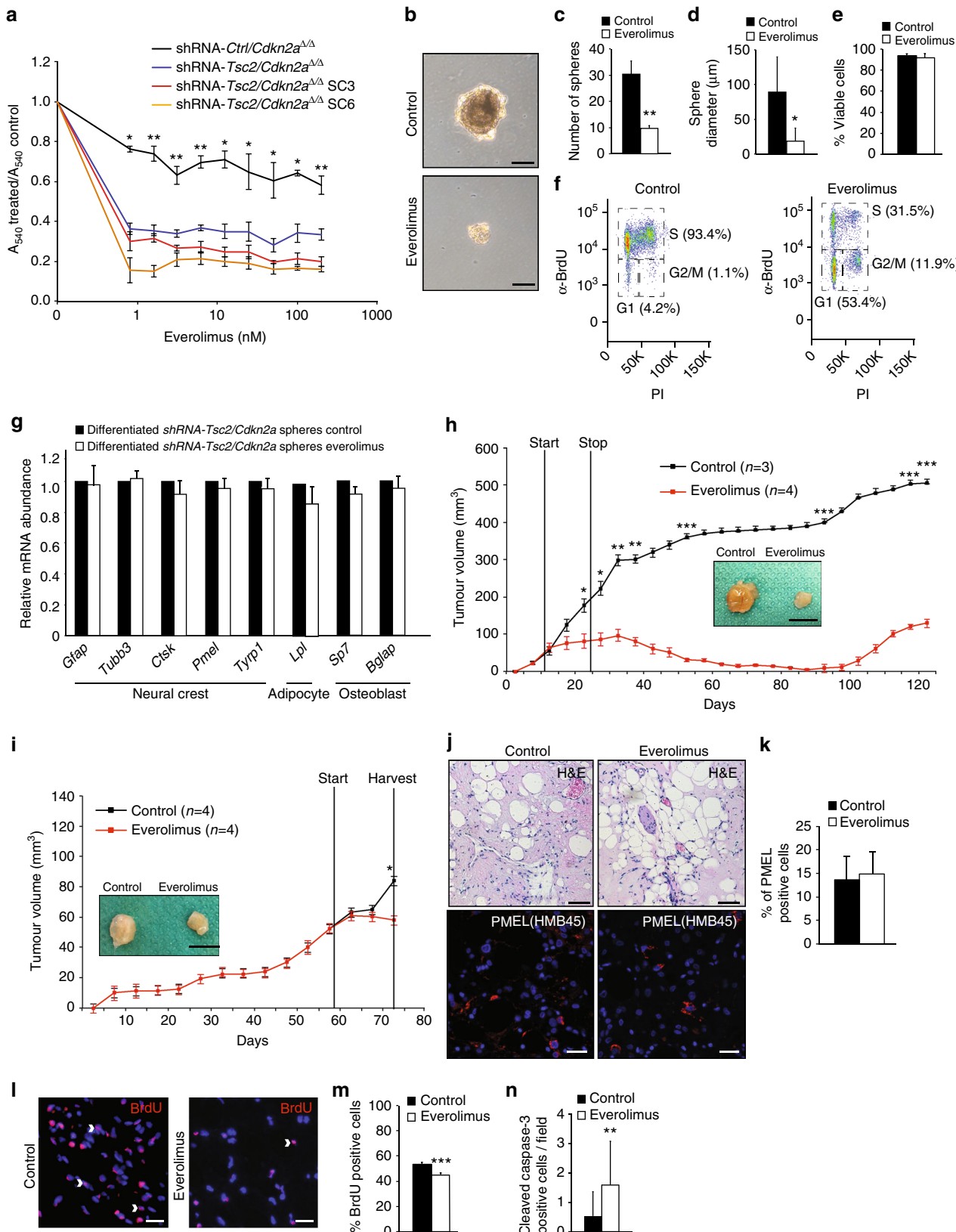

mechanisms operating at the epigenetic, translational or protein levels may permit escape from senescence in human renal AML. In addition, our engineered, sub-cutaneously growing tumours do not fully reproduce all of the morphological features and growth patterns seen in human renal AMLs growing in the kidney. Nonetheless, our studies of our engineered model systems revealed some new features of human renal AMLs, supporting the relevance of our models. Our engineered cells exhibit a proximal tubule gene expression signature, express proximal tubule-specific proteins and display upregulation of several transcription factors that play diverse roles in development and cellular differentiation programmes. These gene expression signatures and protein expression pattern are also at least partly present in human renal AML tumours and in a cell line derived from a renal AML, arguing that this classic 'mesenchymal' tumour might have an epithelial origin.

While unexpected, these findings are potentially consistent with a series of recent studies showing that renal epithelial cells can exhibit a range of different differentiation capacities. As part of the process of repair of injured tubules in response to kidney damage in vivo, renal proximal tubular epithelial cells adopt a mesenchymal appearance and express several molecular markers of mesenchymal cells and of stem cells, including the glycosylated form of CD133, CD24 and high aldehyde dehydrogenase activity[40–43]. Isolation of these cells revealed that they grow as spheres and that they express a strong 'stemness' transcriptional profile[43]. While not yet proven for cells isolated from the proximal tubule, isolation of a similar population of CD133/CD24-expressing parietal epithelial cells from the Bowman's capsule of the glomerulus revealed that these cells could be differentiated in culture into several types of renal epithelial cells, adipocytes, osteoblasts, endothelial cells and neuronal cells, suggestive of a multi-potent progenitor cell phenotype[44]. Thus, at least some and potentially all renal epithelial cells appear to exhibit a degree of cellular plasticity and can adopt progenitor or stem cell properties under certain circumstances[45]. Our observations that renal epithelium-specific heterozygous and homozygous deletion of Tsc1 in the mouse kidney causes the emergence of lesions in which at least some cells express molecular markers of renal AML is consistent with the notion that epithelial cells in vivo might have the capacity to become transformed to generate renal AMLs. Our genetically engineered tumour models provide the first experimental tools to further explore this idea and to define the molecular pathways that govern renal AML tumour biology, for example to assess the potential functions of upregulated transcription factors in causing cellular transformation and stemness. Our studies on the mechanistic basis of everolimus therapy provide proof-of-principle that these experimental models might also facilitate the future identification and pre-clinical validation of new therapeutic approaches for renal AMLs.

An interesting unanswered question is why mutation of TSC1 or TSC2 in humans predominantly causes the development of renal AMLs, renal cysts and more rarely of renal carcinomas, whereas Tsc1 or Tsc2 mutation in rats or mice causes renal cysts, adenomas and carcinomas, but does not lead to the formation of renal AMLs[21]. It is possible that the strong phenotypic effect of Tsc1 or Tsc2 deletion in causing epithelial hyperproliferation in the mouse kidney means that AMLs do not have the necessary time to develop in mouse models. Alternatively, we speculate that our cell culture setting in some way reproduces specific micro-environmental conditions, or niches, that are abundant in human kidneys but rare in mouse kidneys in vivo, and that these specific conditions are necessary for the formation of AMLs. One possibility could be the apparent enhanced rate of loss and replacement of renal epithelial cells in human nephrons compared to mouse nephrons that is believed to underlie the apparent 'spontaneous' epithelial to mesenchymal transition and acquisition of features of stemness in normal human kidneys but not in rodent kidneys[41]. Finally, it is possible that AML development requires additional, yet-to-be-identified genetic or epigenetic alterations that cooperate with loss of TSC function to promote the transition of an epithelial cell to an AML stem cell.

Renal AML belongs to a broader class of tumours termed PEComas that are named after the common presence of perivascular epithelioid cells (PEC). PEComas also include AMLs that form in other organs, lymphangioleiomyomatosis of the lung, clear-cell sugar tumour of the lung, clear-cell myomelanocytic tumour of the falciform ligament and other rare clear-cell tumours found at other anatomical sites[46]. PEComas represent a biological enigma as they share several cellular and molecular features and frequently involve TSC mutation, but the cell type(s) of origin of all of these tumours remains unknown; there is no known normal anatomical counterpart of the perivascular epithelioid tumour cell. By presenting evidence that suggests that the cell of origin of renal AMLs could be an epithelial cell, our studies suggest a new paradigm in the understanding of the biology of this disease and also suggest that epithelial cells in other organs might potentially represent the cells of origin of other types of PEComas. In view of our results, the classification of renal AML as a PEComa might be a misnomer and the current pathological terminology might need to be reconsidered.

## Methods

**Mouse strains.** The following mouse strains were utilised in this study: *SCID/beige* mutant mice (*C.B-17/CrHsd-Prkdc^scid Lyst^bg-1*) and C57BL/6J^OlaHsd mice were from

---

**Fig. 8** Everolimus induces cell cycle arrest in AML tumour allografts. **a** Sulforhodamine B cell number assay of total populations of $Cdkn2a^{\Delta/\Delta}$ primary kidney cells infected with empty shRNA (shRNA-Ctrl/$Cdkn2a^{\Delta/\Delta}$) or with shRNA targeting Tsc2 (shRNA-Tsc2/$Cdkn2a^{\Delta/\Delta}$) and two single cell-derived sphere clones (shRNA-Tsc2/$Cdkn2a^{\Delta/\Delta}$ SC3 and SC6) treated with everolimus for 7 days. Cells were grown under adherent conditions and values represent $A_{540}$ ratios of treated to untreated cells. **b** Phase contrast images of shRNA-Tsc2/$Cdkn2a^{\Delta/\Delta}$ spheres formed in suspension culture conditions in the presence or absence of everolimus (1.6 nM, 48 h). Scale bars: 50 µm. **c–f** Number **c**, diameter **d**, cellular viability **e** and cell cycle phase flow cytometry analysis **f** (x-axis propidium iodide (PI) staining, y-axis α-BrdU staining) of shRNA-Tsc2/$Cdkn2a^{\Delta/\Delta}$ sphere cells after 48 h everolimus (1.6 nM) treatment. Cells were incubated with BrdU (30 µM) for 12 h prior to harvesting. **g** mRNA abundance of the indicated genes in shRNA-Tsc2/Cdkn2a spheres differentiated in the presence or absence of everolimus (1.6 nM, 48 h). **h, i** Long term **h** and short term **i** growth of shRNA-Tsc2/$Cdkn2a^{\Delta/\Delta}$ AML allograft tumours treated with saline (control) or everolimus (10 mg/kg i.p. daily for 2 weeks). The timepoints of starting and stopping therapy are indicated. Images of tumours from each group at the end points of the two experiments are shown. Scale bar: 1 cm. **j** Representative histological images (H&E staining) and PMEL(HMB45) immunofluorescence stainings of shRNA-Tsc2/$Cdkn2a^{\Delta/\Delta}$ AML-like allograft tumours from the experiment shown in **i**. **k** Percentage of PMEL positive cells in shRNA-Tsc2/$Cdkn2a^{\Delta/\Delta}$ AML-like allograft tumours from the experiment shown in **i**. **l** Immunofluorescence staining using α-BrdU antibody (red) and DAPI staining (blue), arrowheads highlight positive nuclei. Proliferating cells were labelled by injection of mice with BrdU (80 µg/g body weight i.p.) 32, 24 and 8 h prior to harvesting tumours. **m** Quantification of percentage of nuclei in tumours from **i** that were labelled with BrdU. **n** Quantification of number of cells per ×40 microscopy field staining positively for cleaved caspase 3 in tumours from **i**. All graphs depict mean ± s.d. Student's t test, n = 3. *P < 0.05; **P < 0.01; ***P < 0.001

Harlan Laboratories. Cdkn2a[fl/fl] (FVB;129P2-Cdkn2a[tm2Brn]/Cnrm)[47] were from EMMA (EM:00407). Sox10-Cre mice have been previously described[48]. Tsc1[fl/fl] (Tsc1[tm1Djk]/J)[24] mice were from Jackson Laboratory (005680). Ai14 td-Tomato reporter mice (B6.Cg-Gt(ROSA)26Sor[tm14(CAG-tdTomato)Hze]/J)[34] were from Jackson Laboratory (007914). Pax8-rtTA/TRE-LC1[33] mice were obtained from Prof. Carsten Wagner, University of Zurich and were intercrossed with Tsc1[fl/fl] mice. Approximately 6–8 week old were exposed to doxyxycline (5 mg/ml) in drinking water supplemented with 5% sucrose for 5 days. Animals were subsequently kept on normal drinking water for 2 months. All mouse experiments were approved by the Veterinary Office of the Canton of Zurich under the licence 06/2013. No statistical method was used to predetermine sample size. The experiments were not randomised and the investigator was not blinded to the genotype of mice.

**Allograft and xenograft experiments**. For allograft and xenograft experiments, SCID/beige mice were anesthetised and injected subcutaneously with $5 \times 10^6$ cells suspended in 50% Matrigel (BD, no.354230). Tumour growth was monitored throughout the experiment and tumour size measured with calipers every 3 days. Everolimus treatment (10 mg/kg i.p.) of mice harbouring AML allograft tumours was performed daily for 2 weeks.

**Cell culture**. Primary MEFs were isolated from E13.5 embryos from WT (C57BL/6) or Cdkn2a[fl/fl] strains, cultured in DMEM plus 10% fetal bovine serum (FBS) in cell-culture incubators supplemented with 5% $CO_2$ and maintained at 5% $O_2$. Murine primary kidney cells were isolated and cultured in complete K-1 culture medium (DMEM-F12 medium (D6421, Sigma) supplemented with 0.5% FBS, 25 mM Glutamine, 5 mg/ml insulin, 1.25 ng/ml prostagladin E1, 34 pg/ml 3,3,5 Triiodothyronine, 5 mg/ml human Apo-transferrin, 1.75 ng/ml sodium selenite, 20 ng/ml Hydrocortisone and 25 ng/ml murine epidermal growth factor) as described[49]. For non-adherent cultures to induce sphere formation, kidney cells ($2 \times 10^3$) were plated in ultra-low attachment culture plates (Corning, 3471) and incubated in complete K-1 culture medium. The spheroids appeared after ~7–10 days in culture. Spheres were dissociated into single cells using Accutase (Gibco, A-11105–01) for replating or for other assays. Human Renal Proximal Tubular Epithelial Cells (RPTEC) TH1 cell line is derived from primary human renal proximal tubule epithelial cells immortalised by two lentiviral vectors carrying the human telomerase (hTERT) and the SV40 T antigen[35]. These cells, as well as B16F1 cells, were cultured in DMEM plus 10% FBS in cell culture incubators at 37 °C, 5% $CO_2$ and 5% $O_2$. The TRI-102 and TRI-103 cell lines were kindly provided by Elizabeth Henske (Brigham and Women's Hospital, Boston) and in K1 medium in cell culture incubators at 37 °C, 5% $CO_2$ and 5% $O_2$. RPTEC TH1, TRI-102 and TRI-103 cells tested negatively for mycoplasma and were validated based on resistance to selection markers (Neo or Zeo) as well as by western blotting for exogenously expressed proteins (SV40 Large T-antigen).

Adipocyte differentiation: Cells obtained from spheres were cultured in DMEM high Glucose, 10% FBS, supplemented with 0.5 mM IBMX, 0.25 µM Dexamethasone, 1 µg/mL Insulin and 2 µM Rosiglitarone. 48 h later medium was changed to DMEM high Glucose, 10% FBS, supplemented with 1 µg/mL Insulin for 48 h. After this treatment, cells were cultured in DMEM high Glucose, 10% FBS without supplements for 7 days.

Neuralcrest differentiation: Cells obtained from spheres were cultured in DMEM-F12 medium, 5% FBS, supplemented with 160 nM TPA (12-O-tetradecanoyl phorbol-13-acetate), 50 ng/mL mSCF (Stem cell factor, mouse, recombinant), 100 nM ET-3 (Endothelin 3, human), 25 ng/mL bFGF (Fibroblast growth factor-basic, human), 100 ng/mL α-MSH (α-Melanocyte stimulating hormone) and 1 nM Dexamethasone for 10 days.

Osteoblast differentiation: Cells obtained from spheres were cultured in DMEM high Glucose, 10% FBS, supplemented with 0.2 mM ascorbic acid, 0.25 µM Dexamethasone and 10 mM β-Glycerol phosphate for 14 days.

Epithelial differentiation: Cells obtained from spheres were cultured in complete K-1 media for 10 days.

**Viral infections**. MEFs and primary kidney cells were infected with adenoviruses expressing GFP (Vector Biolabs, 1060) or Cre-GFP (Vector Biolabs, 1700), lentiviruses (LKO.1) expressing non-silencing hairpin (Addgene, 10879) or shRNA against Tsc1 (Sigma, TRCN0000238187) or shRNA against Tsc2 (Sigma, TRCN0000306244). For lentiviral-mediated knockdown of Cdkn2a and Tsc2, we generated a MuLEvector[50] containing U6 promoter-driven expression of shRNA against Cdkn2a, 7SK promoter-driven expression of shRNA against Tsc2 and PGK promoter-driven expression of puromycin resistance. Human RPTECs were infected with lentiviruses expressing non-silencing hairpin (Addgene, 1864) or four different shRNAs against TSC2 (Sigma, TRCN0000295896, TRCN0000010455, TRCN000010454 and TRCN000040182). All cells were incubated overnight in virus-containing medium in the presence of 4 µg/ml polybrene (Sigma-Aldrich, no. H9268). Drug selection was performed 48 h after transduction using 4 µg/ml of puromycin. Cells were isolated from dissected tumours by digestion for 1 h at 37 °C with 1 mg/ml collagenase type II (Gibco, Life Technologies), washed twice with PBS, and cultured in DMEM plus 10% FBS.

**Cellular assays**. Cellular senescence was evaluated using the Senescence kit (Calbiochem QIA 117). For proliferation assays, cells were seeded at $2 \times 10^5$ cells per 6 cm dish in triplicate dishes and counted after 3 days before reseeding at the same density for the next passage. All proliferation assays shown are representative of at least three independent experiments. Cell viability within the spheres treated with and without everolimus was measured with a Trypan Blue viability assay.

**Drug Assays**. Transformed primary kidney cells were seeded at a density of $2 \times 10^3$ cells per well on a 96 well plate. After attachment of the cells, everolimus was individually added at different concentrations and medium changed every two days to ensure addition of new drug. Cells were fixed after 7 days in 5% w/v trichloracetic acid and stained in 0.057% w/v Sulforhodamine B (SRB) solution and air-dried. SRB was solubilized by incubation in the 10 mM Tris base solution (pH 10.5) and OD was measured at 540 nm in a micro-plate reader.shRNA-Tsc2/Cdkn2a[Δ/Δ] spheres were cultured in the presence or absence of everolimus (1.6 nM) for 48 h.

**Western blotting, antibody stainings and other stainings**. Western blotting, immunofluorescence or immunohistochemistry were conducted using antibodies against the following proteins or epitopes: β-ACTIN (Sigma-Aldrich, A2228, WB 1:5000), TSC1 (Cell Signaling Technology, 4906 s, WB 1:1000), TSC2 (Cell Signaling Technology, 3990s, WB 1:1000), p16 (Santa Cruz Biotechnology, sc-1207, WB 1:200), p19 (Santa Cruz Biotechnology, sc-32748, WB 1:200), p53 (Novocastra, NCL-p53-CM5p, WB 1:500), HIF-1α (Novus Biologicals, NB100-105, WB 1:500), GLUT1 (Abcam, ab14683, WB 1:2500), ribosomal protein S6 (Cell Signaling Technology 2217 s, WB 1:1000), phospho Ser240/244 ribosomal protein S6 (Cell Signaling Technology, 2215 s, WB 1:1000. IHC 1:200), 4E-BP1 (Cell Signaling Technology, 9644 s, WB 1:1000), phosphoThr37/46 4E-BP1 (Cell Signaling Technology, 2855 s, WB 1:1000), CD31 (Abcam, ab28364, IF 1:100), SMA (Abcam, ab5694, IHC 1.200, IF 1:100), antibody cocktail HMB45 + DT101 + BC199 (Abcam, ab732, IF 1:100), PMEL/gp100(HMB45) (Abcam, ab137078, WB 1:1000, IHC 1:100, ICC 1:250),HMB45 (DAKO A/S, M0634, IHC 1:200), PDGFRβ (Abcam, ab32570, IHC 1:100), VIMENTIN (Cell Signaling Technology, 5741,WB 1:1000, IF 1:200, ICC 1:200), E-CADHERIN (BD Bioscience, 610181, WB 1:1000and Abcam, ab11512, IHC 1:50, ICC 1:100), pan-CYTOKERATIN (BMA Biomedicals AG,T1302,IHC 1:100), GFAP (Abcam, ab7260, ICC 1:5000), NAPI2A[51] (IF 1:100, ICC 1:100), AQP1 (Abcam, ab15080, IF 1:100), THP (Santa Cruz Biotechnology, sc-20631, IF 1:200), CD133(Abnova, PAB12663, IF 1:100, ICC 1:100), CD10 (Novocastra Laboratories Ltd,NCL-L-CD10-270, IHC 1:200), RCC (Ventana-Cell Marque, 760-4273, IHC 1:200),RFP (Bioconcept, 600-401-379, IF 1:200, ICC 1:200), BrdU (Merck Milipore, MAB3510, IF 1:200), cleaved CASPASE-3 (Cell Signaling, 9661, IF 1:400), S100 (Abcam, ab183979, IHC 1:100), NG2 (Abcam, ab129051, IHC 1:100), MELAN-A (Santa Cruz Biotechnology, sc-20032, IF 1:50, ICC 1:50), CATHEPSIN K (Abcam, ab19027, WB 1:1000, IHC 1:200, ICC 1:200), MiTF (Abcam, ab20663, WB 1:1000, ICC 1:200), PAX8 (Protein Tech, 10366-1-AP, IHC 1:800), MYC (Abcam, ab32072, WB 1:1000).Validations of the primary antibodies are provided on the manufacturer's websites or in the referenced citations. Full scans of all western blots presented in this study are shown in Supplementary Fig. 13.

Characterisation of adipocytes was performed using Nile Red (Molecular Probes[TM]) or Oil-Red-O staining. Cells were fixed in 10% formalin at room temperature for 1 h and after washing them in 60% isopropanol and drying them, Oil-Red-O solution (0.5% Oil-Red-O solution) was added for 2 h. Cells were washed with distilled water and counterstained with hematoxylin.

**RNA analyses**. Total RNA was extracted from cells and tissues using NucleoSpin RNA Kit (Machery-Nagel) or RNeasy Mini Kit (Quiagen) and cDNA synthesis was done with random hexamer primers and Ready-To-Go You-Prime First-Strand Beads (GE Healthcare). Primers used for real time PCR analyses are listed in Supplementary Table 1. For deep sequencing, RNA samples were sequenced using an IluminaHiSeq 2000 sequencer at the Functional Genomics Center Zurich (FGCZ). GSEA analyses were conducted using the Molecular Signatures Database (http://software.broadinstitute.org/gsea/msigdb/index.jsp).

**Flow cytometry**. Flow cytometry analysis of the mesenchymal stem cell markers was performed as described in mouse multipotent mesenchymal stromal cell marker antibody panel protocol (R&D systems, SC018). Everolimus treated and untreated sphere cells were incubated with BrdU (30 µM) for 12 h prior to harvesting and cell-cycle analysis by flow cytometrywas conducted as described[52].

**Analyses of human AMLs**. Tissues samples for immunofluorescence and real-time quantitative PCR were obtained at the University Hospital of Zurich (Zurich, Switzerland). The study was approved by the local ethics commission (reference number KEK-ZH-Nr. 2014-0604) with informed patient consent. RNA was extracted from punches of paraffin-embedded renal AML tumours and normal kidney tissue from the same patient.

**Electron microscopy**. Tissue culture samples were fixed in 2.5% buffered glutaraldehyde overnight, postfixed in 1% osmium tetroxide for 2 h, dehydrated in graded alcohols, incubated in pure propylenoxide and then propylenoxide-Epon mix and then embedded in Epon. The sections were cut with diamond knives on an ultramicrotome (Ultracut E, Reichert Jung, 90 nm sections), stained with uranyl acetate and lead citrate, and examined using a Hitachi TEM type H-7650 electron microscope.

**Statistics**. Statistical significance was determined by two-tailed Student's test, with a $P$ value <0.05 considered to be statistically significant.

**Data availability**. All data relevant to this publication are provided in the Figures and Supplementary Information.

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

## Acknowledgements

We are grateful to Helena Fisher for her technical help, to Angela Broggini assistance with electron microscopy and to the Cellular Oxygen and Anaesthesiology groups from Institute of Physiology, University of Zurich for their support in the study. We would like to thank Elizabeth Henske (Brigham and Women's Hospital, Boston, USA) for providing cell lines used in this study. This work was supported by grants to IJF from European Research Council (260316), Swiss Cancer Foundation (KFS-3693-08-2015), Novartis Foundation for medical-biological research (15B095) and to A.F.G. from Swiss National Science Foundation (PMPDP3_164462).

## Author contributions

I.J.F. and A.F.G. designed the study; A.F.G., M.A., S.B., T.H., S.H., O.S. conducted experiments; P.J.W., S.B. conducted pathological analyses; L.S., O.S. provided mouse models and scientific ideas and I.J.F., A.F.G. wrote the manuscript with input from all authors.

## Additional information

**Competing interests:** The authors declare no competing financial interests.

