## [Peer Review File · Nature Communications]

Reviewers' comments:

Reviewer #1 (Remarks to the Author):

Goncalves et al. report a novel model of angiomyolipoma, tumors arising primarily in the kidney of which the cell-of-origin has been a mystery. Angiomyolipomas are a major cause of morbidity in patients with tuberous sclerosis complex (TSC) and also occur sporadically. The lack of a bona fide mouse model of AML has been a major roadblock in the TSC field. This work is of considerable interest. The concept of introducing shRNA for *Cdkn2a* to enhance the survival of *Tsc2*-deficient cells is a major advance, and overall the work is well written and the data are clearly presented. However, there are major concerns about the conclusion that AML arise from epithelial progenitors and about the specificity of the findings for TSC, in addition to other concerns, as detailed below.

1. In Figure 1, since cells were isolated from entire mouse kidney and some vimentin positive cells were present, the mesenchymal origin of the AML allografts cannot conclusively be ruled out.

2. A field of HMB-45 positive cells is shown in Fig 1. This is considered a "gold standard" for AML. However, the percentage of positive cells within the tumor is not provided. A Western showing clear positivity for the HMB-45 antibody is essential, with comparison to a normal kidney lysate, because these antibodies can have non-specific immunoreactivity in mouse tissues.

3. Was MiTF positivity observed in the AML, and if yes, what was the percentage positivity? Was MiTF induced in the cultured cells or the spheroids?

4. The in vitro differentiation in Figure 2 of *shTSC2/Cdkn2a Δ/Δ* cells lack a control, so there is no way to determine whether this differentiation is TSC2 dependent. A comparison to spheres representing a different genotype could be made to attempt to prove that there is TSC2-dependence. Presumably, spheres of many different genotypes could be induced to differentiate using these specific media – is this specific for TSC2.

5. Again, there is no examination of MiTF, another "gold standard" of AML.

6. Cathepsin K is highly expressed by AML cells, and is not examined.

7. The HMB-45/Mart1 staining in 2d is not typical in appearance. A Western blot is absolutely essential.

8. What percentage of the allograft tumors from spheres 2 and 4 were HMB-45 positive? Again, a Western would be very helpful for all the markers

9. How many clonal cell lines were tested? Two are shown in 2e, 4 are included in Fig 3. In general, heterogeneity of the spheres and clonally derived spheres should be better described. Do all the clonally derived sphere cells express PAX8? What is the clonogenic efficiency of the spheres? Were all the single cell derived sphere able to form tumors that expressed AML markers?

10. In Figure 3, what is the expression pattern of MiTF, CatK, and other melanocyte genes? What is the gene expression status of Pax8 in the 4 clones? It would be of interest to study clones that were able to form AML-like tumors, vs. those that were not, and again critical to establish the TSC specificity of these findings. Presence of proximal tubule markers (RCC, CD133, CD10) in the AML allograft does not indicate that this allograft arises from proximal tubule epithelial cells.

11. In Figure 4, lineage tracing experiment with doxycycline inducible Cre expression under the Pax8 promoter crossed with R26 loxSTOPlox TdTomato for permanent labeling of shows that the AML-like tumors are arising from Pax8 expressing cells. However, Pax8 is also involved in kidney development, so it is possible that the AML-like cells arise from Pax8 positive cells prior to epithelial differentiation. The stem cell population might be very small, perhaps less than 1 % of total cell population. This is particularly challenging since kidney epithelium arises from embryonic mesenchyme.

12. It would be good to introduce some quantitative data for the differentiation. including Western Blot analysis, in Fig 4.

13. In Fig 4e, were markers of AML detected such as HMB-45?

14. In Figure 5, the Pax8-Cre-driven inducible deletion of TSC1 in adult renal epithelia using Pax8-rtTA;TRE-LC1;Tsc1f/fl mice develop severe polycystic phenotype with intermixed regions of adenoma like proliferation, but not AML questioning the Pax8 expressing cells as a cell of origin of

AML. The authors observe a very small number of cells that appear to have immunoreactivity with HMB-45 (~35 cells per entire kidney). The immunoreactivity is very atypical; usually HMB-45 has a clumped cytoplasmic staining. The authors conclude, based on this sparse and atypical staining, that this shows that loss of TSC function in renal epithelial cells can cause their differentiation into cells that express molecular markers of renal AML, strongly supporting the physiological relevance of our cell culture-based findings." This statement vastly overstates what can be concluded from these cells.

15. In Figure 6, treatment of the AML-like tumor allografts with Everolimus decreases tumor size, but the histology looks very similar in panel I, with no evidence of a change in differentiation. Was this studied more comprehensively, rather than just by visual inspection? Did everolimus decrease HMB-45 immunoreactivity? Again, this should be shown by Western, not only by IHC, because of issues with antibody specificity in the mouse.

16. Does mTOR inhibition alter the differentiation pattern in vitro and in the spheroids?

17. The references to Fig 6h and 6i in the text on page 9 are reversed.

18. Based on the concerns outlined above, the title and abstract should be revised. The statement that "deletion of Tsc1 ... caused differentiation in vivo into cells expressing characteristic AML markers" in the abstract is of particular concern, based on the comments above about the HMB-45 staining. The conclusion that the cell of origin "is an epithelial cell" overstates the data; perhaps "could be an epithelial cell." The response to everolimus adds little to the validation of this model (in contrast to the last sentence of the abstract) since virtually all models of TSC-deficiency demonstrate similar in vivo responses to mTOR inhibitors.

Reviewer #2 (Remarks to the Author):

The work presented in the manuscript is interesting and the argument for angiomyolipoma is forcefully stated. However, histologically, these tumors really do not look like angiomyolipomata. The staining with HMB-45, often used for angiomyolipoma, is seen in some forms of renal cancer, such as the translocation varieties, and these tumors in fact are thought to arise from proximal tubule epithelial cells. This is therefore problematic for narrowing the the interpretation to angiomyolipoma.

The formation of adipose tissue in the model is somewhat interesting, and begs the question if it stains with S100. I bet that it will. The adipose component of an angiomyolipoma does not stain with S100. Oddly, these adipocyte-like cells may have this morphology because of lipid droplet accumulation akin to nonalcoholic fatty liver hepatocytes. Another odd stain that often is positive for angiomyolipomata is NG-2, and the angiomyolipomata cells have very expanded rough endoplasmic reticulum. I wonder if these EM findings would be present in the model, and if so, would it be in response to the Tsc reduction and not indicative of the angiomyolipoma cell, but rather the mTORC1 state.

Another issue is that the cells involved do not histologically resemble angiomyolipoma cells, even the epithelial variety. The cystic lesion cells are dramatically different than human angiomyolipoma cystic cells in humans or mice.

In the absence of a truly in depth comparative characterization and comparison, the merit of the rest of the effort cannot be adequately assessed. The work appears to be of good quality, but what it means is very difficult to understand.

While some similarities in staining and appearance as well as the presence of fat does not make the argument compelling to someone who has worked with angiomyolipoma for a career, the authors may be onto something very interesting, particularly as it relates to renal cell carcinoma. I encourage the authors to explore this avenue.

There is a very rare form of 'renal cell carcinoma- not otherwise specified' in the TSC patient population. The epithelial cell finding and the staining make me think that the authors may have novel insight into this process.

Responses to Reviewer 1.

We are pleased to submit a majorly revised version of our manuscript entitled "Evidence that renal angiomyolipoma neoplastic stem cells arise from renal epithelial cells". We are most grateful for the helpful and insightful input. We now provide an extensive series of new analyses that we believe fully address all of the concerns that were raised by the reviewers. This has led to a vast improvement of our study and a further strengthening of our conclusions. Our point-by-point responses to each of the issues raised are listed below, the changes that have been made to the manuscript are shown in red text in the manuscript document and the following list highlights all of the new data that has been added in the new submission.

Figure 1f: Addition of new staining

Figure 2e: Addition of new staining

Figure 3d: Addition of new stainings

Figure 5a: Addition of analyses of a new cohort of aged mice

Figure 5b: Addition of new stainings

Figure 5c: Addition of new high magnification images

Figure 7: New figure based on analysis of a human renal AML cell line

Figure 8g: Analysis of cellular differentiation in cell culture

Figure 8j,k: Analysis of cellular staining frequency in tumour allografts under therapy

Suppl. Fig 3: New analyses (Electron microscopy, staining quantification, mRNA analysis, western analysis)

Suppl. Fig 4: New analyses (new antibody stainings, description of another tumour model as negative control)

Suppl. Fig. 5: New analyses (stainings with new markers).

Suppl. Fig. 6: New analyses (analyses of two new sets of negative controls via stainings, mRNA expression and western blot).

Suppl. Fig. 7d,e: New western blot analyses and addition of data about non-tumour forming clones

Suppl. Fig. 9: New molecular analyses by western, mRNA and staining

Suppl. Fig. 10e-g: New stainings, western blotting and quantification of stainings

Suppl. Fig. 11: New description of a new cohort of aged mice

Suppl. Fig. 13: New figure with full blots of all western blots shown in the manuscript

Reviewer #1 (Remarks to the Author):

Goncalves et al. report a novel model of angiomyolipoma, tumors arising primarily in the kidney of which the cell-of-origin has been a mystery. Angiomyolipomas are a major cause of morbidity in patients with tuberous sclerosis complex (TSC) and also occur sporadically. The lack of a bona fide mouse model of AML has been a major roadblock in the TSC field. This work is of considerable interest. The concept of introducing shRNA for *Cdkn2a* to enhance the survival of *Tsc2*-deficient cells is a major advance, and overall the work is well written and the data are clearly presented. However, there are major concerns about the conclusion that AML arise from epithelial progenitors and about the specificity of the findings for TSC, in addition to other concerns, as detailed below.

1. In Figure 1, since cells were isolated from entire mouse kidney and some vimentin positive cells were present, the mesenchymal origin of the AML allografts cannot conclusively be ruled out.

After plating the primary kidney cells we could observe by immunofluorescence and western blot that the majority of cells express E-CADHERIN and only a small percentage express VIMENTIN. To exclude the mesenchymal origin of our allografts we performed lineage tracing experiments that clearly prove that the cell of origin of our model system is arenal epithelial cell (Fig. 4 and Supplementary Fig. 10).

2. A field of HMB-45 positive cells is shown in Fig 1. This is considered a “gold standard” for AML. However, the percentage of positive cells within the tumor is not provided.

A Western showing clear positivity for the HMB-45 antibody is essential, with comparison to a normal kidney lysate, because these antibodies can have non-specific immunoreactivity in mouse tissues.

Regarding the issue of immunoreactivity of the PMEL(HMB45) antibodies used in the data that we initially presented, we would like to clarify that we used two different antibodies, an antibody cocktail HMB45+DT101+BC199 (Abcam, ab732) which are mouse antibodies and PMEL/gp100(HMB45) (Abcam, ab137078) which is a rabbit antibody. These two stainings showed similar results. In the revised manuscript we quantified the percentage of PMEL positive cells in allograft tumors (Supplementary Fig. 3e,f). We also now demonstrate expression in allografts by western blotting in comparison to a wild type kidney lysate and B16F1 lysate (melanoma cell line) as a positive control (Supplementary Fig. 3h). We also now demonstrate upregulation of *Pmel* mRNA expression in allografts (Supplementary Fig. 3g).

3. Was MiTF positivity observed in the AML, and if yes, what was the percentage positivity?

Was MiTF induced in the cultured cells or the spheroids?

Unfortunately we were unable to reliably optimize MiTF staining in formalin fixed tissue and therefore we could not stain our mouse allografts with it to calculate the percentage of MiTF positive cells. However, we now show by western blotting that MiTF is expressed in allografts when compared to kidney cells (Supplementary Fig. 7d). Additionally we now show the increased expression in allografts of *Tyrp1* mRNA, (the melanocyte-specific gene product involved in melanin synthesis, which is regulated by MiTF), as well as increased expression of another melanocyte gene, *Mlana* (also known

as MART-1) (Supplementary Fig. 3g). In Supplementary Fig. 9a we demonstrate that MiTF is detectable by western blotting in spheres and allografts and in Supplementary Fig. 9b we demonstrate that *Tyrp1* mRNA expression is upregulated in spheres and allografts.

4. The in vitro differentiation in Figure 2 of shTSC2/*Cdkn2a*^{ΔΔ} cells lack a control, so there is no way to determine whether this differentiation is TSC2 dependent. A comparison to spheres representing a different genotype could be made to attempt to prove that there is TSC2-dependence. Presumably, spheres of many different genotypes could be induced to differentiate using these specific media – is this specific for TSC2.

Thank you for raising this issue. In the revised version of the manuscript we now include two different negative controls in the differentiation assays. Firstly, we compare cells with *Tsc2/Cdkn2a* double knockdown to *Cdkn2a* single knockdown and show that the differentiation only occurs in the double knockdown cells (see Supplementary Fig. 6a-c), emphasizing that the phenotype of multiple differentiation is dependent on loss of *Tsc2* function.

We also present data to address the second issue regarding whether the AML tumour forming phenotype and cell culture differentiation phenotypes are truly related to the *Tsc2/Cdkn2a* genotype or whether any genotype that causes sphere formation would give similar results. While we are unaware of any other studies that have described similar findings of generation of spheres (there are very few laboratories that work with primary kidney cells in the context of studying tumour formation), by chance, our laboratory has been working on other tumour types and in the context of one of these studies we identified (unpublished data) that overexpression of the *Myc* oncogene together with knockout of the *Trp53* tumour suppressor gene also induces sphere formation in primary renal epithelial cells and allows the formation of allograft tumours. In the new version of the manuscript we include a description of this experimental system (Supplementary Fig. 4d-g) and show that the allograft tumours have a completely different morphology to the AML-like tumours and that the *Myc/Trp53* cells cannot be differentiated into the different lineages that the *Tsc2/Cdkn2a* cells can be differentiated into (Supplementary Fig. 6a-c). This further emphasizes the specificity of our phenotypes. We have also used these *Myc/Trp53* cells, spheres and tumours as negative controls in several other experiments in the new manuscript.

5. Again, there is no examination of MiTF, another “gold standard” of AML. Cathepsin K is highly expressed by AML cells, and is not examined.

While our MiTF antibodies could not detect MiTF in formalin fixed tissue, they do work in cultured cells and in western blotting. We now demonstrate MiTF expression in spheres (Fig. 3d – and also show MART-1/MELAN A expression in these spheres). We show MiTF and MART-1 expression in neural crest differentiated spheres (Supplementary Fig. 5) along with a melanoma cell line as a positive staining control, and we show MiTF expression in allograft and spheres by western blotting (Supplementary Fig. 9a)

We also now present a series of data (essentially in every experimental setting described in the

paper) where we have investigated CATHEPSIN K expression by immunostaining, mRNA expression and western blotting. These data show that cell, spheres and allografts express CATHEPSIN K. These data are presented in Fig. 1f, Fig. 2e, Fig. 3d, Fig. 5b, Fig. 7d, Fig. 8g, Supplementary Fig 3g,i, Supplementary Fig. 4c, Supplementary Fig. 5, Supplementary Fig. 6b, Supplementary Fig. 7c,d, Supplementary Fig. 9a, and Supplementary Fig. 10e.

7. The HMB-45/Mart1 staining in 2d is not typical in appearance. A Western blot is absolutely essential.

We repeated the staining with the PMEL/gp100 (HMB45) antibody (Abcam, ab137078) and now present better images in various places in the figures (eg. in Fig. 2d, Fig. 5c) showing the predominantly cytoplasmic staining. We would also like to point out that the stainings shown in Supplementary Fig. 12d were obtained at the Department of Pathology, University of Zurich using the staining protocol that is used for their routine diagnostics. We also now show western blots of PMEL protein in Supplementary Figs. 3h, 6c, 7d and 10f.

8. What percentage of the allograft tumors from spheres 2 and 4 were HMB-45 positive? Again, a Western would be very helpful for all the markers

The percentage of allograft tumors from Sphere clone 2 and 4 was calculated as requested (Supplementary Fig. 3f) and these percentages are similar to the percentage of positive cells in allografts from total sphere populations. A western blot is also provided in Supplementary Fig. 7d showing the expression of PMEL, MiTF and CATHEPSIN K in shRNA-Tsc2/Cdkn2a^{ΔΔ} SC2 and SC4 in comparison to WT kidney lysate and B16F1 melanoma cells as positive control.

9. How many clonal cell lines were tested? Two are shown in 2e, 4 are included in Fig 3.

We tested 6 clonal lines, of which 4 (SC2, SC3, SC4, SC6) formed allograft tumours and were able to be differentiated to express the different cell lineage markers. The other 2 clonal cell lines (SC1 and SC5) did not form tumours and did not possess the full differentiation capacity. These lines are now described in the text and shown in Supplementary Fig. 7e.

In general, heterogeneity of the spheres and clonally derived spheres should be better described.

We have now included a comment in the text relating to this issue of potential clonal heterogeneity in spheres.

Do all the clonally derived sphere cells express PAX8?

We tested this question and yes, all allograft tumours derived from the 4 tumour-forming cell lines express PAX8, consistent with an epithelial origin or phenotype. This new data is presented in Supplementary Fig. 9c.

What is the clonogenic efficiency of the spheres? Were all the single cell derived sphere able to form tumors that expressed AML markers?

See comments above.

10. In Figure 3, what is the expression pattern of MITF, CatK, and other melanocyte genes?

In Fig. 3d we now provide stainings showing that PMEL, MITF, CATHEPSIN K and MART-1 are expressed in spheres. In the new Supplementary Fig. 3g we demonstrate increased relative mRNA abundance of *Pmel*, *Tyrp1*, *Mlana* and *Ctsk* in shRNA-Tsc2/Cdkn2a^{ΔΔ} allograft, shRNA-Tsc2/Cdkn2a^{ΔΔ} SC2 allograft and shRNA-Tsc2/Cdkn2a^{ΔΔ} SC4 allograft in comparison to WT kidney.

What is the gene expression status of Pax8 in the 4 clones?

See above

It would be of interest to study clones that were able to form AML-like tumors, vs. those that were not, and again critical to establish the TSC specificity of these findings.

As described above we present the data about the 2 clones that did not form allograft tumours in the new manuscript. We agree that this may be one possible avenue for future study to gain more insight into the mechanisms of aberrant cellular differentiation in renal AML tumours, but these studies are beyond the scope of the current study.

Presence of proximal tubule markers (RCC, CD133, CD10) in the AML allograft does not indicate that this allograft arises from proximal tubule epithelial cells.

We agree with this statement. However, when taking all of our findings together, based on reverse-engineered mouse cells, reverse-engineered human cells, mouse genetics and our new studies of a renal AML cell line, our observations of mRNA expression signatures of the proximal tubule as well as expression of proteins that are specifically found in renal proximal tubule cells, combined with the very strong data related to lineage tracing (see comments below), we believe that these data are at least highly consistent with a potential proximal tubule origin. We agree that there is theoretically a possibility that another cell type could adopt these proximal tubule features (although our lineage tracing strongly argues against this). We have altered the strength of our statements and conclusions in various places in the manuscript to reflect this possibility.

11. In Figure 4, lineage tracing experiment with doxycycline inducible Cre expression under the Pax8 promoter crossed with R26 loxSTOPlox TdTomato for permanent labeling of shows that the AML-like tumors are arising from Pax8 expressing cells. However, Pax8 is also involved in kidney development, so it is possible that the AML-like cells arise from Pax8 positive cells prior to epithelial differentiation. The stem cell population might be very small, perhaps less than 1 % of total cell population. This is particularly challenging since kidney epithelium arises from embryonic mesenchyme.

It is important to highlight that Cre-recombination (induced by feeding doxycycline) was induced in adult mice (6-8 weeks of age), long after kidney development has taken place, meaning that the concerns about a complication related to PAX8 expression in the embryonic mesenchyme are unfounded. PAX8-driven Cre activity in our experimental setting labels adult renal epithelial cells.

In the new manuscript we provide an additional powerful piece of evidence that further validates the relevance of our engineered models and provides further evidence of involvement of the proximal tubule. This is described in the following copy/paste text from the new manuscript.

We next obtained a set of isogenic renal AML cell lines to investigate whether these cells display similarities to our engineered models. TRI-102 cells are an immortalised (E6/E7 and hTERT) derivative of a *TSC2* null primary cell culture that was derived from a human renal AML³⁵. TRI-103 is a derivative of TRI-102 into which *TSC2* expression has been re-introduced³⁵. Similarly to our engineered cells, TRI-102 but not TRI-103 cells formed spheres when grown in epithelial medium on non-adherent plates (Fig. 7a). When grown as adherent cultures, both TRI-102 and TRI-103 expressed the mesenchymal marker VIMENTIN as well as the epithelial marker E-CADHERIN (Fig. 7b). TRI-102 cells also expressed the proximal tubule marker proteins CD133 and NAPI2A and the expression of these proteins was absent or reduced in TRI-103 cells (Fig. 7b). TRI-102 cells when grown as adherent cells or as spheres expressed higher mRNA abundance than TRI-103 of the transcription factors and some of the proximal tubule genes that we identified in our engineered systems (Fig. 7c). Finally, TRI-102 but not TRI-103 cells could be differentiated into Nile Red, CATHEPSIN K and PMEL positive cells in appropriate media (Fig. 7d). These findings establish that a cell line derived from a human renal AML have similar renal proximal tubule epithelial features and differentiation capacities to our engineered cellular systems.

12. It would be good to introduce some quantitative data for the differentiation. including Western Blot analysis, in Fig 4.

Using real time PCR assays to provide a quantitative measure of cellular differentiation, we now conducted side-by-side differentiation assays of sphere-selected cells derived from shRNA-*Tsc2+Cdkn2a* infected kidney cells from *Pax8-rtTA;TRE-LC1;R26R-TdTomato* mice together with two negative control genotypes (shRNA-*Cdkn2a* and *Myc/Trp53^{ΔΔ}*) and show that only the differentiated shRNA-*Tsc2+Cdkn2a* cells express the lineage specific genes, validating the differentiation capacity of spheres derived from the lineage-tracing genetic background. We also further analysed the differentiation status of the allograft tumours using immunohistochemical staining as described below. All of these analyses show no differences between the spheres and tumours from this genetic background to the spheres and tumours derived from C57Bl/6 mice.

13. In Fig 4e, were markers of AML detected such as HMB-45?

We now present stainings of PMEL, CATHEPSIN K, PDGFR β , SMA and VIMENTIN (Supplementary Fig. 10e) and quantification of the percentage of PMEL positive cells (Supplementary Fig. 10e).

14. In Figure 5, the Pax8-Cre-driven inducible deletion of *TSC1* in adult renal epithelia using *Pax8-rtTA;TRE-LC1;Tsc1fl/fl* mice develop severe polycystic phenotype with intermixed regions of adenoma like proliferation, but not AML questioning the Pax8 expressing cells as a cell of origin of AML. The authors observe a very small number of cells that appear to have immunoreactivity with HMB-45 (~35 cells per entire kidney). The immunoreactivity is very atypical; usually HMB-45 has a clumped cytoplasmic staining. The authors conclude, based on this sparse and atypical staining, that this shows that loss of TSC function in renal epithelial cells can cause their differentiation into cells

that express molecular markers of renal AML, strongly supporting the physiological relevance of our cell culture-based findings.” This statement vastly overstates what can be concluded from these cells.

We repeated the PMEL staining and present new higher magnification images showing the predominantly cytoplasmic staining pattern (Fig. 5c). We have also added new data showing CATHEPSIN K staining (Fig. 5b). Most importantly, we have added an entirely new set of analyses of aged *Tsc1* heterozygous mice and altered our conclusion. The following is a copy/paste of the text in the new manuscript:

Given that the strong hyperproliferative phenotype of *Tsc1* homozygous deletion prevented aging experiments, we generated a cohort of 12 *Pax8-rtTA;TRE-LC1;Tsc1^{fl/+}* heterozygous mutant mice and analysed them 1 year after inducing gene deletion in 6-8 week old mice. By histological sectioning of kidneys we identified 20 cysts (14 were lined by a single layer of epithelial cells, 6 cysts displayed atypical, multilayered morphology), 5 small tumours with epithelial morphology and 28 lesions comprising closely packed cells with low cytoplasmic volume, often occurring close to blood vessels (Supplementary Fig. 11). These lesions exhibit a similar morphology to lesions that were identified upon *Tsc1* deletion under a ubiquitous promoter and that were proposed to represent potential precursors of renal AMLs²¹. Consistent with this idea, the lesions in our mice harboured many cells that were positive for phosphorylation of ribosomal protein S6 (Ser240/244) and some cells that were positive for CATHEPSIN K, SMA, and more rarely for PMEL (Fig. 5b). Thus, the epithelial-specific PAX8-Cre driver appears to recreate a similar phenotype to that previously described with the ubiquitous Cre driver. These observations provide *in vivo* evidence to show that loss of TSC function in renal epithelial cells can cause the formation of different lesions that contain cells that express several different molecular markers of renal AML, consistent with our *ex vivo* cell engineering studies.

15. In Figure 6, treatment of the AML-like tumor allografts with Everolimus decreases tumor size, but the histology looks very similar in panel I, with no evidence of a change in differentiation. Was this studied more comprehensively, rather than just by visual inspection? Did everolimus decrease HMB-45 immunoreactivity? Again, this should be shown by Western, not only by IHC, because of issues with antibody specificity in the mouse.

We agree that there were no obvious differences in the appearance of the treated and untreated tumours. We conducted PMEL immunostainings of control and everolimus-treated allograft tumours and we could show that there is no difference in the percentage of PMEL positive cells between control and everolimus allograft tumors (Fig 8j,k).

16. Does mTOR inhibition alter the differentiation pattern in vitro and in the spheroids?

To test the idea that cellular differentiation might be altered by everolimus we compared the differentiation of sphere cells using different differentiation media in the presence or absence of everolimus and used real time PCR to analyse specific lineage target genes (Fig. 8g). These assays showed no effect on differentiation.

17. The references to Fig 6h and 6i in the text on page 9 are reversed.

Thank you, this has been corrected in the new manuscript.

18. Based on the concerns outlined above, the title and abstract should be revised. The statement that “deletion of Tsc1 ... caused differentiation in vivo into cells expressing characteristic AML markers” in the abstract is of particular concern, based on the comments above about the HMB-45 staining. The conclusion that the cell of origin “is an epithelial cell” overstates the data; perhaps “could be an epithelial cell.” The response to everolimus adds little to the validation of this model (in contrast to the last sentence of the abstract) since virtually all models of TSC-deficiency demonstrate similar in vivo responses to mTOR inhibitors.

Thank you for these comments. We agree and have accordingly altered the title, abstract and several of the conclusions.

Responses to Reviewer 2.

We are pleased to submit a majorly revised version of our manuscript entitled "Evidence that renal angiomyolipoma neoplastic stem cells arise from renal epithelial cells". We are most grateful for the helpful and insightful input. We now provide an extensive series of new analyses that we believe fully address all of the concerns that were raised by the reviewers. This has led to a vast improvement of our study and a further strengthening of our conclusions. Our point-by-point responses to each of the issues raised are listed below, the changes that have been made to the manuscript are shown in red text in the manuscript document and the following list highlights all of the new data that has been added in the new submission.

Figure 1f: Addition of new staining

Figure 2e: Addition of new staining

Figure 3d: Addition of new stainings

Figure 5a: Addition of analyses of a new cohort of aged mice

Figure 5b: Addition of new stainings

Figure 5c: Addition of new high magnification images

Figure 7: New figure based on analysis of a human renal AML cell line

Figure 8g: Analysis of cellular differentiation in cell culture

Figure 8j,k: Analysis of cellular staining frequency in tumour allografts under therapy

Suppl. Fig 3: New analyses (Electron microscopy, staining quantification, mRNA analysis, western analysis)

Suppl. Fig 4: New analyses (new antibody stainings, description of another tumour model as negative control)

Suppl. Fig. 5: New analyses (stainings with new markers).

Suppl. Fig. 6: New analyses (analyses of two new sets of negative controls via stainings, mRNA expression and western blot).

Suppl. Fig. 7d,e: New western blot analyses and addition of data about non-tumour forming clones

Suppl. Fig. 9: New molecular analyses by western, mRNA and staining

Suppl. Fig. 10e-g: New stainings, western blotting and quantification of stainings

Suppl. Fig. 11: New description of a new cohort of aged mice

Suppl. Fig. 13: New figure with full blots of all western blots shown in the manuscript

Reviewer #2 (Remarks to the Author):

The work presented in the manuscript is interesting and the argument for angiomyolipoma is forcefully stated. However, histologically, these tumors really do not look like angiomyolipomata. The staining with HMB-45, often used for angiomyolipoma, is seen in some forms of renal cancer, such as the translocation varieties, and these tumors in fact are thought to arise from proximal tubule epithelial cells. This is therefore problematic for narrowing the interpretation to angiomyolipoma.

We agree with the statement of the reviewer that our engineered tumours histologically do not look exactly like human renal angiomyolipoma. In the new manuscript we take care to highlight the many histological and molecular similarities between our numerous engineered models analysed *ex vivo* and *in vivo* that are all consistent with an angiomyolipoma-like phenotype. We now also add additional data based on renal epithelial-specific heterozygous knockout of *Tsc1* knockout mice and we add an analysis of a human renal AML cell line – both of which provide even more lines of evidence that are consistent with our proposal. We have strengthened our molecular analyses by including more markers of renal AML in various experimental settings (we have added protein and mRNA data pertaining to CATHEPSIN K, MiTF, MART-1/MELAN A, TYRP1, S100 and NG2 expression). This data is also consistent with our original conclusions. We have nonetheless “softened” some of our conclusions and altered the title, abstract and discussion to reflect the fact that we have generated “models” that we refer to in the manuscript as giving AML-like allograft tumours.

The formation of adipose tissue in the model is somewhat interesting, and begs the question if it stains with S100. I bet that it will. The adipose component of an angiomyolipoma does not stain with S100. Oddly, these adipocyte-like cells may have this morphology because of lipid droplet accumulation akin to nonalcoholic fatty liver hepatocytes.

While mature normal adipocytes stain positively for S100, adipocytes in human renal AMLs typically do not express S100, but other non-adipocyte components of the tumour have been reported to stain positively in about 28% of renal AML cases²⁸. Consistent with this, in our allograft tumours, adipocytes stained negatively for S100 and spindle shaped cells stained positively (Fig. 1f and Supplementary Fig. 4b). This result represents another molecular consistency between our model and human renal AML. Our lineage tracing studies presented in Fig. 4 are also consistent with the adipocytes in the engineered allograft model arising from renal epithelial cells.

Another odd stain that often is positive for angiomyolipomata is NG-2,

We now show that NG2 is expressed in our mouse allograft tumours (Supplementary Fig. 4b) but not in another tumour genotype (*Myc/Trp53^{ΔΔ}*) that serves as a negative control.

and the angiomyolipomata cells have very expanded rough endoplasmic reticulum. I wonder if these EM findings would be present in the model, and if so, would it be in response to the Tsc reduction and not indicative of the angiomyolipoma cell, but rather the mTORC1 state.

To address this query we performed electron microscopy on our engineered mouse allograft models and indeed observed an expanded rough endoplasmic reticulum (Supplementary Fig. 3a-d). This observation represents another similarity between our engineered model and human renal AML.

Another issue is that the cells involved do not histologically resemble angiomyolipoma cells, even the epithelial variety. The cystic lesion cells are dramatically different than human angiomyolipoma cystic cells in humans or mice. In the absence of a truly in depth comparative characterization and comparison, the merit of the rest of the effort cannot be adequately assessed. The work appears to be of good quality, but what it means is very difficult to understand.

We now provide a much more in depth characterization of our engineered allograft tumours, which has highlighted a number of additional similarities to human renal AML. We would also like to stress that the histological appearance of our engineered models should not be considered in isolation and it is in fact essential to consider all of the other work that we describe in the manuscript as we present pieces of evidence that come from many different lines of experimental investigation that are all mutually consistent in strengthening our argument. (see also answer to questions below).

While some similarities in staining and appearance as well as the presence of fat does not make the argument compelling to someone who has worked with angiomyolipoma for a career, the authors may be onto something very interesting, particularly as it relates to renal cell carcinoma. I encourage the authors to explore this avenue. There is a very rare form of 'renal cell carcinoma- not otherwise specified' in the TSC patient population. The epithelial cell finding and the staining make me think that the authors may have novel insight.

We agree that while there are many similarities between our models and human renal AML at the molecular and histological levels, there are also differences - such as the frequency and appearance of the epithelial cells. It should be pointed out that there are no autochthonous mouse models of renal AML, nor human renal AML xenograft models, meaning that it is impossible to know exactly how an allograft or xenograft renal AML tumour is expected to look. In light of the fact that there are no other animal models, we believe that the many similarities to the human disease seen in our models therefore make our study exciting and novel.

We also acknowledge that we have generated "experimental models" that mimic several aspects of renal angiomyolipoma, rather than entirely reproducing the exact tumour in the kidney in a mouse. This remains an open challenge for the field. As we additionally point out in the revised discussion:

"We show that loss of function of TSC2 in genetic backgrounds that abrogate senescence and proliferation arrest is sufficient to convert mouse and human renal proximal tubular epithelial cells into neoplastic stem cells that give rise to allograft or xenograft tumours that reproduce many of the histological and molecular features of human renal AMLs, providing the first models of AML-like tumours. It should be noted that these tumours represent "models" as the genetic backgrounds that we employed in this study to overcome cellular senescence induced by loss of TSC2 function are apparently not reflected in the genetics of human renal AMLs, for example, genetic mutations or deletions of *TP53*, *CDKN2A* or other senescence regulating tumour suppressor genes have not been observed in AML tumours. It remains possible that yet-to-be-identified mechanisms operating at the epigenetic, translational or protein levels may permit escape from senescence in human renal AML. In addition, our engineered, sub-cutaneously growing tumours do not fully reproduce all of the morphological features and growth patterns seen in human renal AMLs growing in the kidney. Nonetheless, our studies of our engineered model systems revealed some new features of human renal AMLs, supporting the relevance of our models. Our engineered cells exhibit a proximal tubule gene expression signature, express proximal tubule-specific proteins and display upregulation of several transcription factors that play diverse roles in development and cellular differentiation programmes. These gene expression signatures and protein expression pattern are also at least partly present in human renal AML tumours and in a cell line derived from a renal AML, arguing that

this classic “mesenchymal” tumour might have an epithelial origin.”

While we present many lines of independent evidence that are consistent with our models representing models of AML (histomorphology, patterns of immunoreactivity for numerous marker proteins, mRNA expression patterns, similarities to human renal AML and renal AML cell line), we see no histological resemblance of our allograft tumours with the very rare unclassifiable subset of renal cell carcinomas that arise in TSC patients in which (commonly cuboidal) tumour cells grow in a variety of architectural patterns including tubules/papillae, compact nests or small tubules or sheets (Yang et al. *Am J Surg Pathol.*, 2014, 38:895-909).

REVIEWERS' COMMENTS:

Reviewer #1 (Remarks to the Author):

Gonzales et al.

The authors have done remarkably comprehensive experimentation to address virtually every concern, often with multiple parallel approaches. The manuscript is greatly improved and represents a very important advance in the pathogenesis of angiomyolipoma, with therapeutic implications. The inclusion of human angiomyolipoma-derived cells is an additional strength.

Yu et al., *Am J Physiol*, 2003 should be included as a reference (in addition to, or instead of Ref #35) regarding the derivation and genetic validation of the angiomyolipoma cells.